# High-throughput mapping of the phage resistance landscape in *E. coli*

Vivek K. Mutalik[1,2]*, Benjamin A. Adler[2,3], Harneet S. Rishi[4,5], Denish Piya[2,3], Crystal Zhong[1], Britt Koskella[6], Elizabeth M. Kutter[7], Richard Calendar[8], Pavel S. Novichkov[1], Morgan N. Price[1], Adam M. Deutschbauer[1,2,9], Adam P. Arkin[1,2,3,4,5]*

1 Environmental Genomics and Systems Biology Division, Lawrence Berkeley National Laboratory, Berkeley, California, United States of America, 2 Innovative Genomics Institute, Berkeley, California, United States of America, 3 Department of Bioengineering, University of California – Berkeley, Berkeley, California, United States of America, 4 Biophysics Graduate Group, University of California – Berkeley, Berkeley, California, United States of America, 5 Designated Emphasis Program in Computational and Genomic Biology, University of California – Berkeley, Berkeley, California, United States of America, 6 Department of Integrative Biology, University of California – Berkeley, Berkeley, California, United States of America, 7 The Evergreen State College, Olympia, Washington, United States of America, 8 Department of Molecular and Cell Biology, University of California – Berkeley, Berkeley, California, United States of America, 9 Department of Plant and Microbial Biology, University of California – Berkeley, Berkeley, California, United States of America

* vkmutalik@lbl.gov (VKM); aparkin@lbl.gov (APA)

**Data Availability Statement:** Sequencing data have been uploaded to the Sequence Read Archive under BioProject accession number PRJNA645443 [https://www.ncbi.nlm.nih.gov/bioproject/PRJNA645443/]. Supplementary Tables with

## Abstract

Bacteriophages (phages) are critical players in the dynamics and function of microbial communities and drive processes as diverse as global biogeochemical cycles and human health. Phages tend to be predators finely tuned to attack specific hosts, even down to the strain level, which in turn defend themselves using an array of mechanisms. However, to date, efforts to rapidly and comprehensively identify bacterial host factors important in phage infection and resistance have yet to be fully realized. Here, we globally map the host genetic determinants involved in resistance to 14 phylogenetically diverse double-stranded DNA phages using two model *Escherichia coli* strains (K-12 and BL21) with known sequence divergence to demonstrate strain-specific differences. Using genome-wide loss-of-function and gain-of-function genetic technologies, we are able to confirm previously described phage receptors as well as uncover a number of previously unknown host factors that confer resistance to one or more of these phages. We uncover differences in resistance factors that strongly align with the susceptibility of K-12 and BL21 to specific phage. We also identify both phage-specific mechanisms, such as the unexpected role of cyclic-di-GMP in host sensitivity to phage N4, and more generic defenses, such as the overproduction of colanic acid capsular polysaccharide that defends against a wide array of phages. Our results indicate that host responses to phages can occur via diverse cellular mechanisms. Our systematic and high-throughput genetic workflow to characterize phage-host interaction determinants can be extended to diverse bacteria to generate datasets that allow predictive models of how phage-mediated selection will shape bacterial phenotype and evolution. The results of this study and future efforts to map the phage resistance landscape will lead to new insights into the coevolution of hosts and their phage, which can

complete CRISPRi data are deposited here: https://doi.org/10.6084/m9.figshare.11859216.v1. In addition, the complete data from RB-TnSeq experiments are deposited here: https://doi.org/10.6084/m9.figshare.11413128. And Dub-seq experiments are deposited here: https://doi.org/10.6084/m9.figshare.11838879.v2. The underlying data for all figures are provided in supporting information file S1 Data.

**Funding:** This project was funded by the Microbiology Program of the Innovative Genomics Institute, Berkeley (to VKM, AMD, and APA). The initial concepts for this project were funded by ENIGMA, a Scientific Focus Area Program at the Lawrence Berkeley National Laboratory, supported by the US Department of Energy, Office of Science, Office of Biological and Environmental Research under contract DE-AC02-05CH11231 (to VKM, AMD, and APA). The funders had no role in study design, data collection and analysis, decision to publish, or preparation of the manuscript.

**Competing interests:** I have read the journal's policy and the authors of this manuscript have the following competing interests: VKM, AMD, and APA consult for and hold equity in Felix Biotechnology, Inc.

**Abbreviations:** cAMP, cyclic-AMP; c-di-GMP, cyclic di-GMP; CRISPRi, CRISPR interference; DGC, diguanylate cyclase; dsDNA, double-stranded DNA; Dub-seq, dual-barcoded shotgun expression library sequencing; EOP, efficiency of plating; EPS, exopolysaccharide; FDR, false discovery rate; GlmS, glucosamine-6-phosphate synthase; GOF, gain-of-function; LB, lysogeny broth; LOF, loss-of-function; LPS, lipopolysaccharide; MOI, multiplicity of infection; PDE, phosphodiesterase; RB-TnSeq, random barcode transposon site sequencing; R-M, restriction-modification; RNA-seq, RNA sequencing; sgRNA, single-guide RNA.

ultimately be used to design better phage therapeutic treatments and tools for precision microbiome engineering.

## Introduction

Bacterial viruses, or bacteriophages (phages), are obligate parasites that infect specific bacterial strains. Phages represent the most abundant biological entities on earth and are key ecological drivers of microbial community dynamics, activity, and adaptation, thereby impacting environmental nutrient cycles, agricultural output, and human and animal health [1–8]. Despite nearly a century of pioneering molecular work, the mechanistic insights into phage specificity for a given host, infection pathways, and the breadth of bacterial responses to different phages have largely focused on a handful of individual bacterium-phage systems [9–13]. Bacterial sensitivity/resistance to phages is typically characterized using phenotypic methods such as cross-infection patterns against a panel of phages [14–27] or by whole-genome sequencing of phage-resistant mutants [28–33]. As such, our understanding of bacterial resistance mechanisms against phages remains limited, and the field is therefore in need of improved methods to characterize phage-host interactions, determine the generality and diversity of phage resistance mechanisms in nature, and identify the degree of specificity for each bacterial resistance mechanism across diverse phage types [13,25,26,34–48].

Unbiased and comprehensive genetic screens that are easily transferable and scalable to new phage-host combinations would be highly valuable for obtaining a deeper understanding of phage infection pathways and phage resistance phenotypes [49–54]. Such genome-scale studies applied to different phage-host combinations have the unique potential to identify commonalities or differences in phage resistance patterns and mechanisms [18,25,28,55–57]. There have been few attempts to use genetic approaches for studying genome-wide host factors essential in phage infection. These loss-of-function (LOF) genetic screens broadly use bacterial saturation mutagenesis [49,54,58–61] or an arrayed library of single-gene deletion strains for studying phage-host interactions [50,51,53,62,63]. Consequently, these studies have generally involved laborious experiments on relatively few phages and their hosts, and scaling the approach to characterize hundreds of phages is challenging.

To overcome these technological limitations, we have developed three high-throughput genetic technologies that enable fast, economical, and quantitative genome-wide screens for gene function, which are suitable for discovering host genes critical for phage infection and bacterial resistance. Random barcode transposon site sequencing (RB-TnSeq) allows genome-wide insertion mutagenesis leading to LOF mutations [64]; a pooled CRISPR interference (CRISPRi) approach, which allows partial inhibition of gene function via transcriptional inhibition [65]; and dual-barcoded shotgun expression library sequencing (Dub-seq) [66], which queries the effects of gene overexpression. All three technologies can be assayed across many conditions at low cost, as RB-TnSeq and Dub-seq use randomized DNA barcodes to assay strain abundance (BarSeq [67]), whereas quantification of the pooled CRISPRi strains only requires deep sequencing of the guide sequences.

In RB-TnSeq, genome-wide transposon insertion mutant libraries labeled with unique DNA barcodes are generated, and next-generation sequencing methods are used to map the transposon insertions and DNA barcode at loci in genomes. Although RB-TnSeq can be applied on a large scale across multiple bacteria through barcode sequencing [68], it is limited to nonessential genes. Partial LOF assays such as CRISPRi use a catalytic null mutant of the

Cas9 protein (dCas9) guided by chimeric single-guide RNA (sgRNA) to programmably knock down gene expression, thereby allowing the probing of all genes (including essential genes) and more precise targeting of intergenic regions [65]. We have recently applied CRISPRi technology to systematically query the importance of approximately 13,000 genomic features of *E. coli* in different conditions [69]. CRISPRi technology has been extended to different organisms to study essential genes [70–78] and has been recently applied to *E. coli* to uncover host factors involved in T4, 186, and λ phage infection [52]. Lastly, Dub-seq uses shotgun cloning of randomly sheared DNA fragments of a host genome on a dual-barcoded replicable plasmid and next-generation sequencing to map the barcodes to the cloned genomic regions. The barcode association with genomic fragments and genes contained on those fragments enables a parallelized gain-of-function (GOF) screen, as demonstrated in our recent work [66]. In contrast to LOF genetic screens, GOF screens to study gene dosage effects on phage resistance have not been broadly reported, except for a recent work on T7 phage using an *E. coli* single-gene overexpression library [51]. There are indications that enhanced gene dosage can be an effective way to search for dominant-negative mutants, antisense RNAs, or other regulatory genes that may block phage growth cycle [9,10,13,51,56,79]. Such GOF screens, when applied in a high-throughput format across diverse phages, may yield novel mechanisms of phage resistance that LOF screens may not uncover.

In this study, we employ these three technologies (RB-TnSeq, Dub-seq, and CRISPRi) as a demonstration of "portable" and "scalable" platforms for rapidly probing phage-host interaction mechanisms. Using *E. coli* K-12 strain and 14 diverse double-stranded DNA (dsDNA) phages, we show that our screens successfully identify known receptors and other host factors important in infection pathways, and they also yield additional novel loci that contribute to phage resistance. We validate some of these new findings by deleting or overexpressing individual genes and quantifying fitness in the presence of phage. Additionally, we used RB-TnSeq and Dub-seq to compare similarity and distinctiveness in phage resistance displayed by *E. coli* strains K-12 and BL21. The comparison of two historical lineages of *E. coli* allowed us to examine how strain-level divergence of genotype can lead to differential susceptibility in phage resistance. Finally, we discuss the implications and extensibility of our approaches and findings to other bacteria-phage combinations and how these datasets can provide a foundation for understanding phage ecology and engineering phage for therapeutic applications.

## Results and discussion

### Mapping genetic determinants of phage resistance using high-throughput LOF and GOF methods

Despite *E. coli* being a well-studied model organism [80,81], there are significant knowledge gaps regarding gene function [82] and phage interaction mechanisms [26,27,47,83,84]. Different serotypes of *E. coli* are also important pathogens with significant global threat and are crucial players in specific human-relevant ecologies [85–87], leading to the question of whether strain variation is also important in predicting the response to phage-mediated selection or whether the mechanisms are likely to be conserved. Such mechanistic information is unavailable for not only pathogenic *E. coli* strains but also for widely used laboratory nonpathogenic strains such as *E. coli* K-12 and BL21 [88–90]. Historically, both *E. coli* K-12 and B (ancestor of BL21) strains have been used in disparate phage studies and have provided foundational knowledge on phage physiology and growth [10,91–93], though phage studies on *E. coli* BL21 are limited relative to those in K-12. Both K-12 BW25113 and BL21 lack functional CRISPR machinery and type I restriction-modification (R-M) system that function as common antiphage systems, and both strains are unique in their genomic content, physiology, and growth

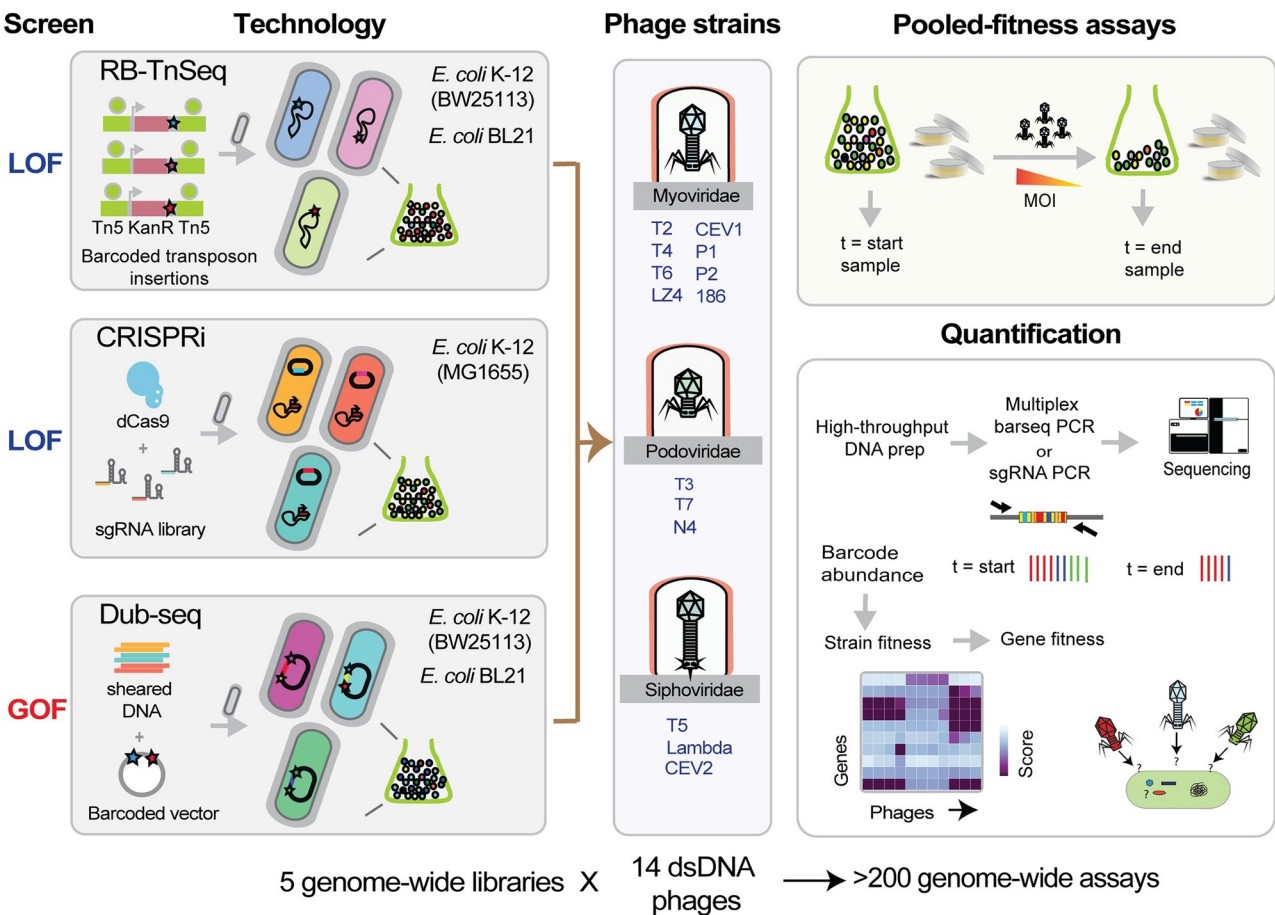

**Fig 1. Overview of high-throughput genome-wide screens.** We used barcoded LOF technologies (RB-TnSeq and CRISPRi) and a GOF technology (Dub-seq) in *E. coli* K-12 (BW25113 and MG1655) to screen for host factors important in phage infection and resistance. In *E. coli* BL21, we performed RB-TnSeq and Dub-seq (but not CRISPRi). We sourced 14 diverse *E. coli* phages with dsDNA genomes, belonging to Myoviridae, Podoviridae, and Siphoviridae families, and performed pooled fitness screens in both planktonic and solid agar formats. Disruption or overexpression of certain genes provide fitness to host in the presence of phages, and we monitor these changes by quantifying the abundance of the DNA barcode or sgRNA associated with each strain. The individual strain abundances are then converted to gene fitness scores (normalized log2 change in the abundance of mutants in that gene). CRISPRi, CRISPR interference; dsDNA, double-stranded DNA; Dub-seq, dual-barcoded shotgun expression library sequencing; GOF, gain-of-function; LOF, loss-of-function; MOI, multiplicity of infection; RB-TnSeq, random barcode transposon site sequencing; sgRNA, single-guide RNA.

characteristics [88–90], thereby serving as a valuable reference for comparing closely related host responses to the same phage selection pressures. To demonstrate the efficacy of our approaches to characterize phage resistance mechanisms and to compare their similarity and differences between two closely related laboratory model strains, we first applied high-throughput genetic screens to *E. coli* K-12 strains (BW25113 and MG1655) and then extended it to *E. coli* BL21 strain (Fig 1).

We used a previously constructed *E. coli* K-12 BW25113 RB-TnSeq library [64] and defined an *E. coli* K-12 MG1655 CRISPRi library [69]. To study host gene dosage and overexpression effects on phage resistance, we used a prior reported GOF Dub-seq library of *E. coli* K-12 BW25113 [66]. We sourced 14 diverse *E. coli* phages with dsDNA genomes, belonging to Myoviridae, Podoviridae, and Siphoviridae families (within the order Caudovirales) (Fig 1). These phages include 11 canonical and well-studied coliphages, each having overlapping but distinct mechanisms of host recognition, entry, replication, and host lysis [94], and two recently reported phages (CEV1 and CEV2), which are known to kill pathogenic shiga toxin–

producing *E. coli* (STEC, O157:H7), and one novel coliphage (LZ4). These 14 phages include T-series phages (T2, T3, T4, T5, T6, T7), N4, 186, λcI857, P1*vir*, P2, and novel isolates of T-like phages (T6-like LZ4, STEC infecting T4-like CEV1 and T5-like CEV2).

The barcoded LOF or GOF libraries were grown competitively in a single-pot assay and placed under phage-mediated selection. Any variation in fitness resulting from disruption or overexpression of specific genes in the presence of phage is therefore exposed to selection, allowing for evolutionary change within the population. We monitor these changes by quantifying the abundance of the DNA barcode associated with each strain. The individual strain abundances are then converted to gene fitness scores, which we define as the normalized log2 change in the abundance of mutants in that gene (Methods). A positive gene fitness score ("positive hit") indicates the strain with deletion or overexpression of a gene realized an increase in relative fitness in the presence of a particular phage (i.e., strains with these genetic changes are more resistant to the phage and are enriched in our phage selection assay). A positive fitness score for a gene in our LOF assay indicates that its encoded product (for example, a phage receptor) is needed for successful phage infection, whereas a positive fitness score for a gene in our GOF assay indicates that its encoded product (for example, a repressor of phage receptor) prevents the phage infection cycle. Negative fitness values, which suggest reduced relative fitness, indicate that the gene(s) disruption or overexpression results in these strains being more sensitive to the phage than the typical strain in the library. Lastly, fitness scores near zero indicate no fitness change for the mutated or overexpressed gene under the assayed condition. Because of the strong selection of phage infection, we anticipated (and indeed observed) that genetically modified strains in our libraries with resistance to a phage would lead to very strong positive fitness values. Although these very strong positive phenotypes are readily interpretable, one consequence is that strains with relatively poor or neutral fitness scores will be swept from the population. Thus, intermediate resistance factors to phage infection will have similar low or negative fitness values as a neutral mutant. As such, most of our focus in this study is on the strong positive fitness scores (Methods). Challenges in identifying intermediate phage-resistant mutants in the presence of highly resistant phage receptor mutations are well appreciated in the field [95].

## RB-TnSeq identifies known receptors and host factors for all 14 phages

To perform genome-wide transposon-based LOF assays, we recovered a frozen aliquot of the *E. coli* K-12 RB-TnSeq library in lysogeny broth (LB [96]) to mid-log phase, collected a cell pellet for the "start," and used the remaining cells to inoculate an LB culture supplemented with different dilutions of a phage in SM buffer. After 8 hr of phage infection in planktonic cultures, we collected the surviving phage-resistant strains or "end" samples (Methods). We also repeated these fitness assays on solid media by plating the library post phage adsorption, incubating the plates overnight, and collecting all surviving phage-resistant colonies. We hypothesized that, given the spatial structure and possibility of phage refuges, fitness experiments on solid media might provide a less stringent selection environment than in liquid pooled assays, such that less fit survivors could potentially be detected. For all assays, recovered genomic DNA from surviving strains was used as templates to PCR amplify the barcodes for sequencing (Methods). The strain fitness and gene fitness scores were then calculated as previously described [64].

In total, we performed 68 RB-TnSeq assays across 14 phages at varying multiplicity of infection (MOI) and 9 no-phage control assays (Methods). For planktonic assays, the gene fitness scores were reproducible across different phage MOIs (Fig 2A), with a median pairwise correlation of 0.90. Because of stronger positive selection in the presence of phages (relative to our

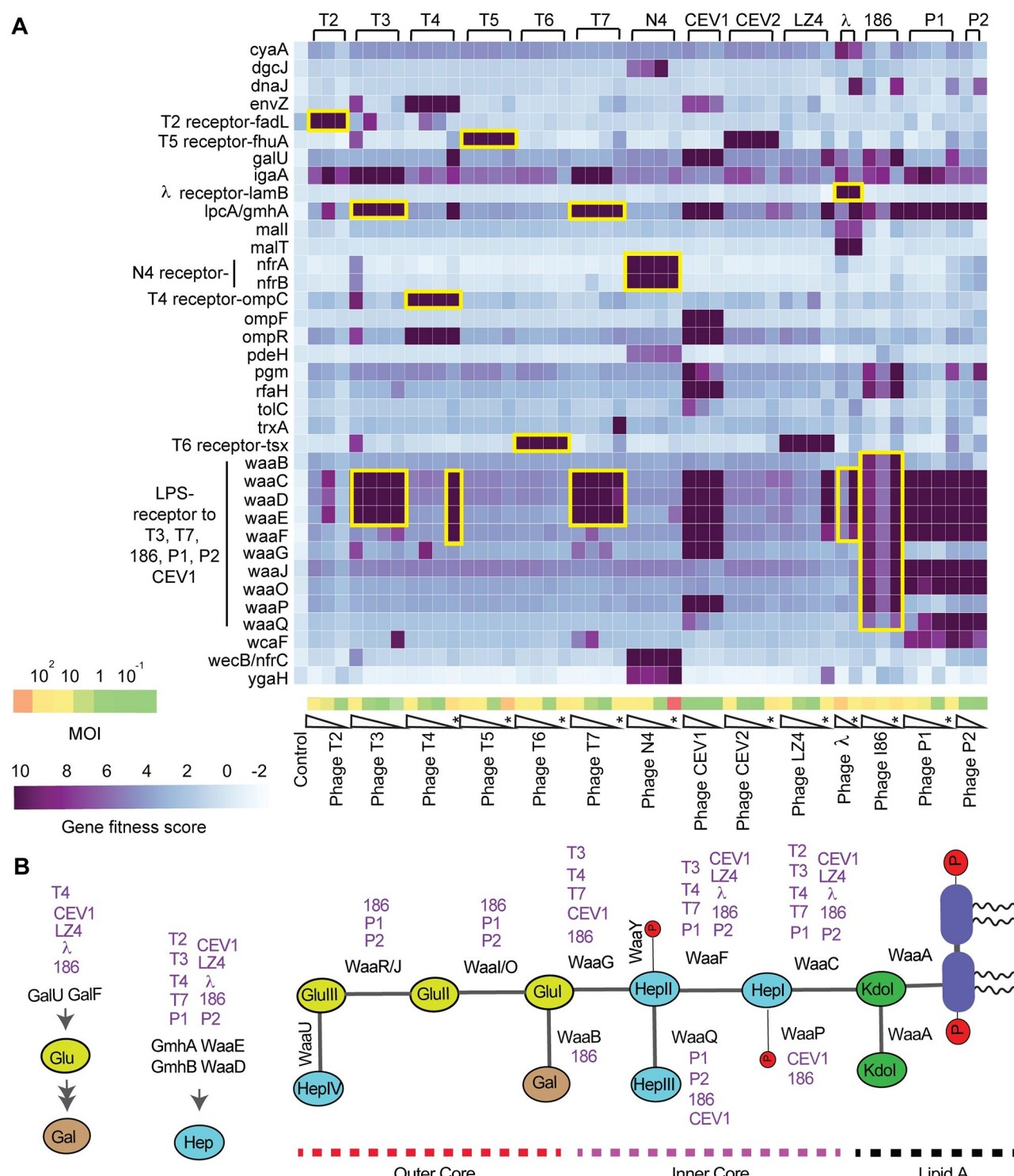

**Fig 2. Heatmap of *E. coli* K-12 RB-TnSeq data for 14 dsDNA phages at different MOI.** (A) Top 36 genes with high-confidence effects and a gene fitness score of ≥6.5 in at least one phage assay are shown. The pooled fitness assays performed on solid agar plates are shown as stars. Yellow boxes highlight genes that encode known receptors for the marked phages. The underlying data for this figure can be found in S1 Data. (B) Schematic of *E. coli* K-12 LPS structure with associated enzymes involved in LPS core biosynthesis. Top-scoring candidates in the presence of a particular phage (at any MOI) are highlighted in purple by associating each enzymatic step with phages. dsDNA, double-stranded DNA; LPS, lipopolysaccharide; MOI, multiplicity of infection; RB-TnSeq, random barcode transposon site sequencing.

typical RB-TnSeq fitness assays with stress compounds and in defined growth media [64,68]), to determine reliable effects across experiments we used stringent filters (gene fitness score ≥ 5, t-like statistic ≥ 5, and estimated standard error ≤ 2) (Methods). Across all replicate experiments, we identified 354 high-scoring hits, which represent 52 unique genes and 133 unique gene-phage combinations (some genes were linked with more than one phage) (S1 Table). In all phage experiments, there was at least one gene with a high-confidence effect. In the presence of some phages (for example, T5 and T6), we observed enrichment of strains with disruption in only one gene (encoding the phage receptor), whereas other phages (such as 186 and λ cI857 phages) showed enrichment with disruptions in multiple genes (Fig 2A).

Phage infection initiates through an interaction of the phage with any bacterial cell surface–exposed molecules ("receptors"), which could be lipopolysaccharide (LPS) moieties, peptido-glycan, teichoic acids, capsules, and proteinaceous components. Any changes in levels or structure of these receptors can compromise efficient phage infection, thereby leading to an improvement in host fitness in the presence of phages. Consequently, receptor mutants are the most commonly found candidates that display phage resistance [49,50,53,58,95,97]. To confirm the validity of our approach, we looked for receptors recognized by many of the canonical phages used in this study for which there are published data available [50,51,91–93,98–101]. Indeed, we found high fitness scores (fitness score > 10, corresponding to >1,000-fold enrichment of transposon mutants) for multiple known phage receptors (Fig 2, S1–S3 Tables). These included genes coding for protein receptors such as *fadL* (long-chain fatty acid transporter) for T2, *ompC* (outer membrane porin C) for T4, *fhuA* (ferrichrome outer membrane transporter) for T5, *tsx* (nucleoside-specific transporter) for T6, *nfrA* and *nfrB* (unknown function) for N4, and *lamB* (maltose outer membrane transporter) for λ. *fhuA* showed high fitness scores in the presence of CEV2 phage, indicating CEV2 has a similar receptor requirement as T5 [102]. We find *tsx* as the top scorer in the presence of novel LZ4 phage, thus appearing to have similar receptor requirement as T6 phage [103]. In addition to protein receptors, we also identified a few high-scoring genes that are known to interfere or regulate the expression of receptors, thereby impacting phage infection. For example, deletion of the EnvZ/OmpR two-component system involved in the regulation of *ompC* and gene products involved in regulation of *lamB* expression (*cyaA*, *malI*, *malT*) all show high fitness scores in the presence of T4 and λ phages, respectively (Fig 2A).

For phages utilizing LPS as their receptors, we found top scores for gene mutations within the *waa* gene cluster, which codes for enzymes involved in LPS core biosynthesis (Fig 2). For example, *waaC*, *waaD*, *waaE*, and *lpcA/gmhA* were the top scorers for T3 and T7 phages, whereas *waaC*, *waaD*, *waaE*, *waaF*, *waaJ*, *waaO*, *waaQ*, and *lpcA/gmhA* showed high fitness in the presence of P1, P2, and 186 phages (Fig 2A and 2B). Even though P1 and P2 phages have been studied for decades, the host factors important in their infection cycle are not fully characterized [93,104]. Our results show that all LPS core components are essential for an efficient P1, P2, and 186 phage infection. CEV1 phage seems to require both outer membrane porin OmpF and LPS core for efficient infection. In addition to *ompF* and its regulator the *envZ/ompR* two-component signaling system, a number of genes involved in the LPS core biosynthesis pathway (*waaC*, *waaD*, *waaE*, *waaF*, *waaG*, *waaP*, *galU*, and *lpcA/gmhA*) and a regulator of genes involved in biosynthesis, assembly, and export of LPS core (*rfaH*) all showed high fitness scores (>10) in the presence of CEV1 phage. Among other top-scoring hits, genes encoding a putative L-valine exporter subunit (*ygaH*) and a diguanylate cyclase (DGC), *dgcJ*, showed stronger fitness in the presence of N4 phage. Both YgaH and DgcJ were not previously known to be involved in N4 phage resistance. Finally, *igaA/yrfF*, which encodes a negative regulator of the Rcs phosphorelay pathway, shows strong fitness scores against eight phages, indicating its importance to general phage resistance. Though *igaA* is an essential gene in *E. coli* [105], our

RB-TnSeq library contained 9 disruptions in *igaA*'s cytoplasmic domain, and these strains seem to tolerate the disruption (S1 Fig). It is known that the Rcs signal transduction pathway functions as an envelope stress response system that monitors cell surface composition and regulates a large number of genes involved in diverse functions including colanic acid synthesis and biofilm formation [106].

Compared to assays performed in planktonic cultures, a few additional gene mutants showed stronger fitness effects on plate assays. For example, *trxA*, encoding thioredoxin 1, is known to be essential for T7 phage propagation and scored high in the solid plate assays but not in our planktonic growth assays (Fig 2). Thioredoxin 1 is a processivity factor for T7 RNA polymerase, and it is reported that T7 phage can bind and lyse a *trxA* deletion strain, though T7 phage propagation is severely compromised [107]. Similarly, we observed higher fitness scores for *galU* in the presence of T4, P1, and λ phages, and *dnaJ* in the presence of 186, P1, and λ phages, on plates but not in broth. GalU catalyzes the synthesis of UDP-D-glucose, a central precursor for synthesis of cell envelope components, including LPS core, and it is known that the growth of P1 phage is compromised on a *galU* mutant [104]. *dnaJ* codes for a chaperone protein and *dnaJ* insertion mutants are known to inhibit the growth of λ phage [108–110]. We also observed higher fitness scores for a number of genes involved in LPS biosynthesis (*waaC*, *waaD*, *waaE*, *waaF*, *lpcA*) in the presence of T4, LZ4, and λ phages when grown on solid plates. These results suggest that LPS components either play an important role in an efficient phage infection cycle or these LPS truncations lead to a destabilized membrane and probably decrease outer membrane protein levels via envelope stress response [106,111–114]. A detailed description about many of the genes we identified in this study is provided in S1 Text. In summary, our results indicate how the abiotic environment can have an important influence on the host fitness and susceptibility and the type of resistance mechanism selected in the presence of different phages [115–119]. While this manuscript was under review, a new work using transposon sequencing to screen receptors for T2, T4, T6, and T7 phages came out [120]. Our results are largely in agreement with this new study, with a few differences that point to difference in the mutant library size (17,100 insertions in 3,253 bacterial genes compared to our RB-TnSeq library with 152,018 barcoded insertions in 3,728 genes).

Overall, our RB-TnSeq LOF screen provided a number of top hits that agree with earlier reports and also yielded a set of novel genes whose role in phage infection was not previously known (S1–S3 Tables). Later in this manuscript, we describe follow-up experiments with 18 of these top-scoring hits from the RB-TnSeq screen to validate their role in phage resistance.

## A CRISPRi screen provides a deeper view of phage resistance determinants

We next employed a rationally designed *E. coli* K-12 MG1655 genome-wide CRISPRi library approach to look for bacterial essential and nonessential genes and genomic regulatory regions important in phage infection and to provide a complementary genetic screen to RB-TnSeq. This CRISPRi library comprises multiple sgRNAs targeting annotated genes, promoter regions, and transcription factor binding sites, a total of 13,000 target regions [69] (Methods). By directing dCas9 to different genomic regions via unique sgRNAs, CRISPRi enables the interrogation of genic and nongenic regions. For the CRISPRi assays, we recovered a frozen aliquot of the library, which was back diluted in fresh media, supplemented with dCas9 and gRNA inducers, and mixed with phage. We assayed T2, T3, T4, T5, T6, N4, 186, CEV1, CEV2, LZ4, and λ phages in planktonic cultures at an MOI of 1, recovered survivors after phage treatment for 3 hr, isolated plasmid DNA, and the variable gRNA region was PCR amplified and sequenced (Methods). During these pooled CRISPRi assays, strains that carry sgRNAs targeting a feature on *E. coli* genome important for phage infection (for example, a phage receptor)

will increase in abundance and will have a positive fitness score. In total, we identified 542 genes (including 75 genes of unknown function), 94 promoter regions, and 44 transcription factor binding sites that show high fitness scores across all phages (sgRNAs with log2 fold-change ≥2 and false discovery rate [FDR]-adjusted $p$-value <0.05) (S4 Table, Methods).

To confirm our assay system correctly identifies host factors important in phage infection, we looked for genes that are known to impact phage infection and also are in agreement with high fitness scores in our RB-TnSeq results. Indeed, the top-scoring hits included sgRNAs that target both the genes coding for phage receptors and their promoter and transcription factor binding sites (Fig 3A, S4 Table). Our CRISPRi data also confirmed many of the top-scoring genes uncovered in RB-TnSeq screen (Fig 3A, S3 Table), thus validating the importance of these genes in specific phage infection pathways. For example, *ompF* was the top-scoring gene for CEV1 phage, and *tsx* showed high fitness scores in the presence of LZ4, whereas *fhuA* was the top scorer for phage CEV2. We also found that sgRNAs targeting *dgcJ* or its promoter showed high fitness scores in the presence of N4 phage, thereby confirming the RB-TnSeq data (Fig 2A). In agreement with RB-TnSeq data, we observed that knockdown of *igaA* yields resistance to 10/11 phages we assayed (Fig 3B–3D).

Among disagreements between the RB-TnSeq and CRISPRi datasets, we found that our CRISPRi screen failed to return some of the highest-scoring genes uncovered in RB-TnSeq dataset. These include genes encoding the EnvZ/OmpR two-component system for T4 and CEV1 phages and YgaH and WecB for N4 phage. Here, we note that we have only one sgRNA targeting *ygaH* and no sgRNAs targeting *wecB/nfrC* region in our CRISPRi library. In addition to these genes, the contribution of LPS core biosynthesis genes in phage infection was less clear in our CRISPRi dataset. For example, both OmpF and LPS seem to be crucial for CEV1 infection from our RB-TnSeq dataset (Fig 2A), whereas the CRISPRi screen data showed high fitness score for *ompF* and not for all core LPS biosynthetic genes (Fig 3A). Nevertheless, we find high fitness scores for genes encoding LPS transport system (*lptABC*) and lipid A biosynthesis enzymes (Fig 3B). In summary, these results indicate that we might be missing few candidates in our CRISPRi screen (as compared to RB-TnSeq) because either some genes lack sufficient sgRNA coverage or even minimal expression of these genes is probably enough for phage infection.

One of the key advantages of CRISPRi is the ability to study the contribution of essential genes on phage infection. We found 11 essential genes (*csrA*, *kdsC*, *lptA*, *lptB*, *lptC*, *lpxA*, *lpxC*, *nusG*, *secE*, *yejM*, and *tsf*) that showed broad fitness advantage (fitness score ≥2 in more than one phage assay) when down-regulated (Fig 3B). None of these essential genes are present in our RB-TnSeq library, except for *yejM*, which has transposon insertions in the C-terminal portion (after 5 putative transmembrane helices) of the protein. The putative cardiolipin transporter encoded by *yejM* and its upstream neighbor *yejL* both show enhanced fitness in the presence of T3, T4, T6, CEV1, CEV2, and LZ4 phages in the CRISPRi dataset (Fig 3A and 3B). These genes have not been previously associated with phage resistance. Although the physiological role of cardiolipin is still emerging, it is known that cardiolipins play an important role in outer membrane protein translocation system and membrane biogenesis [121–123]. A recent study showed that decreased cardiolipin levels activate Rcs envelope stress response [123]. Down-regulation of cardiolipin transport probably results in phage resistance via increased colanic acid biosynthesis, but further mechanistic studies are needed.

Although the fitness benefit phenotype conferred by most top-scoring essential genes in the presence of phages is challenging to interpret, some of these hits agree with previous work. For example, it is known that the down-regulation of genes involved in transcription antitermination (*nusB*, *nusG*) and Sec translocon subunit E (*secE*) shows high fitness in the presence of λ phage and is crucial for the phage growth cycle [124–126]. *lptABC*, *kdsC*, and *lpxAC* are

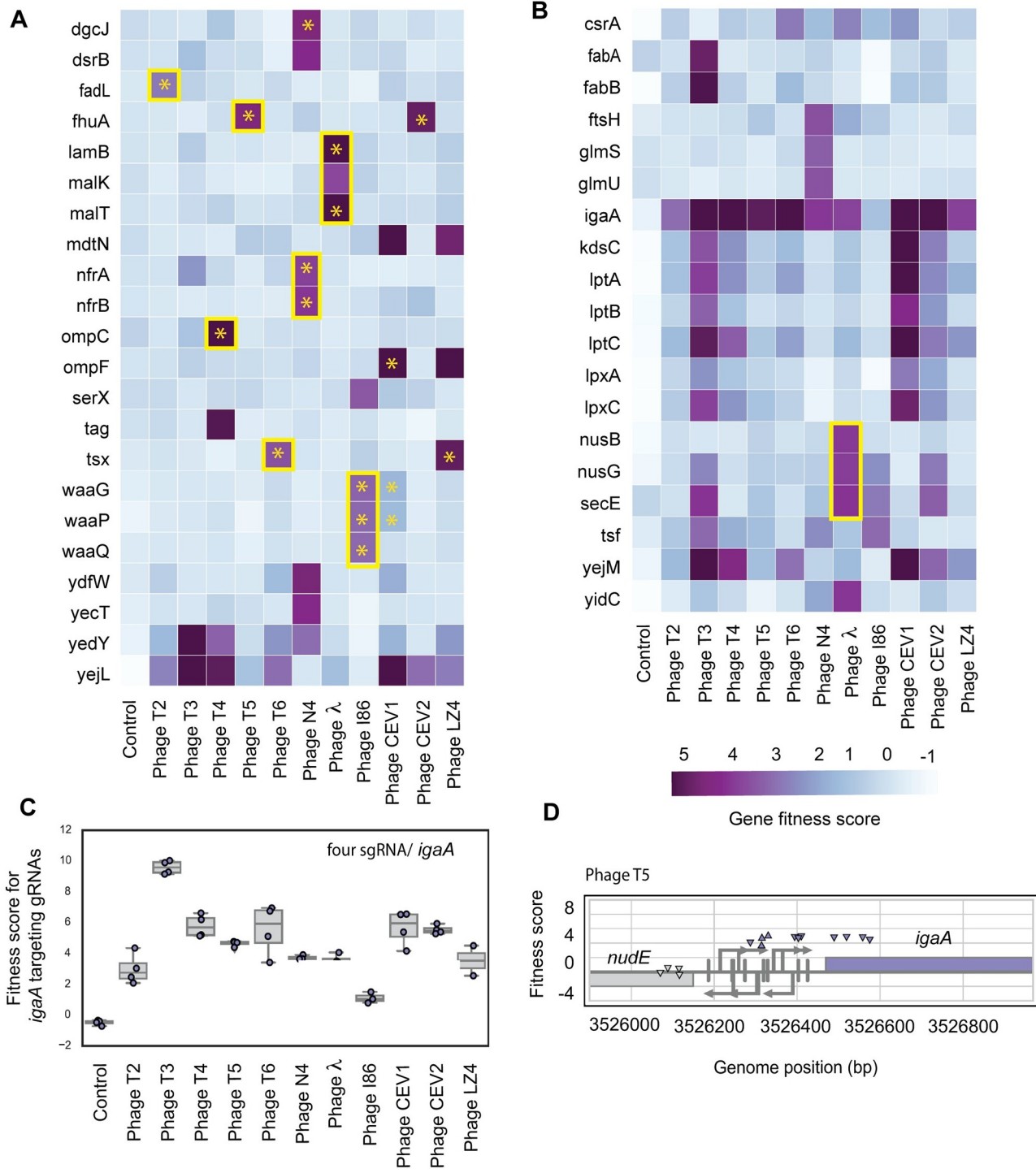

**Fig 3. CRISPRi fitness profiles of *E. coli* nonessential and essential genes in the presence of diverse phages.** (A) Heatmap of top-scoring nonessential genes across 11 phages. Yellow boxes highlight genes that encode phage receptors and are known to interfere with phage growth when down-regulated. Yellow stars indicate these data points are in agreement with RB-TnSeq results. (B) Heatmap of top-scoring essential genes across 11 phages. Yellow boxes highlight genes that are known to interfere with phage growth when down-regulated. (C) Box plot of fitness data for *igaA*-targeting sgRNAs across 11 different phages. Each data point in this plot is a specific sgRNA targeting *igaA*. (D) Genome browser plot for *nudE-igaA* locus with targeting sites for gRNAs and their fitness scores. The downward-facing triangles mean that the sgRNA targeted the nontemplate strand of the gene. Under each promoter, a vertical bar denotes the +1 for the promoter with the stem for the promoter starting at −60 relative to the transcription start site. The underlying data for this figure can be found in S1 Data. CRISPRi CRISPR interference; RB-TnSeq, random barcode transposon site sequencing; sgRNA, single-guide RNA.

known to impact outer membrane biogenesis, LPS synthesis, and transport [113,114,127,128], whereas the RNA binding global regulator CsrA is known to be involved in carbohydrate metabolism and regulation of biofilm [129,130]. Down-regulation of these genes likely leads to pleiotropic effects leading to enhanced fitness in the presence of phages. Our CRISPRi screen also identified a number of *E. coli* tRNA-related genes showing enhanced fitness in the presence of diverse phages (S4 Table). How the down-regulation of host tRNA and tRNA modification genes impacts host fitness, phage growth, and infection cycle is not clear, although recent studies have shown increased aminoacyl-tRNA synthetase activities in the early phage infection cycle [131,132].

Finally, neither the RB-TnSeq nor CRISPRi screens found high scores for *ompF* in the presence of T2 phage and *cmk* in the presence of T7 phage (Figs 2A and 3A). Previous reports had indicated that OmpF serves as a secondary receptor to T2 (in addition to the primary receptor FadL) [133] and Cytidine monophosphate kinase (encoded by *cmk*) is an essential host factor in T7 phage infection [51]. We also did not observe significant fitness scores for host factors that may bias the rate of lysogeny of the two temperate phages (186 and P2), leading to resistance of those lysogens. Our genome-wide screens did not recapitulate these findings probably because these mutants provide an intermediate fitness in the presence of phages [134] and are swept from the population in the presence of highly resistant phage receptor mutations.

In summary, CRISPRi served as a complementary screening technology to RB-TnSeq, validating many RB-TnSeq hits, and provided an avenue to study the role of essential genes and nongenic regions on phage infection.

## Dub-seq enables parallel mapping of multicopy suppressors of phage infection

To study the effect of host factor gene dosage or overexpression on phage resistance, we performed competitive fitness assays using the *E. coli* K-12 Dub-seq library (expressed in the *E. coli* K-12 strain) in the presence of phages. This library is made up of randomly sheared 3-kb genomic DNA of *E. coli* K-12 BW25113 cloned into a dual-barcoded vector with the copy number of approximately 15–20 [66]. Similar to pooled fitness assays with the RB-TnSeq library, we recovered a frozen aliquot of the library to mid-log phase, collected a cell pellet for the initial sample, and used the remaining cells to inoculate an LB culture supplemented with different dilutions of a phage in SM buffer (Methods). Similar to LOF RB-TnSeq assays, after 8 hr of phage infection in planktonic cultures (and overnight incubation in case of plate assays), we collected the surviving phage-resistant strains, isolated plasmid DNA, and sequenced the DNA barcodes (Methods). In total, we performed 67 genome-wide GOF fitness assays in the presence of 13 different phages at varying MOIs, both in planktonic and solid plate assay format and 4 no-phage control experiments (Methods). Here, one genome-wide GOF fitness assay encompasses an ensemble of barcoded vectors with sheared 3-kb genomic region of the whole *E. coli* K-12 BW25113. Overall we identified 233 high-scoring positive hits for the *E. coli* K-12 Dub-seq screen made up of 129 unique phages-gene combinations and found more than five Dub-seq hits per phage that confer positive growth benefit (fitness score ≥ 4, FDR of 0.7, Methods, S5 Table). A positive fitness score for a gene in our Dub-seq assay indicates that the overexpression of that gene interferes with successful phage infection.

The growth benefit phenotypes we observe in Dub-seq assays may not be simply due to overexpression of genes encoded on genomic fragments but rather be due to other potential effects such as increased gene copy number (gene dosage), or other indirect dominant-negative effects may be playing a role [135–140]. For example, overexpression or higher copy number of a regulatory region might lower the effective concentration of a transcription factor

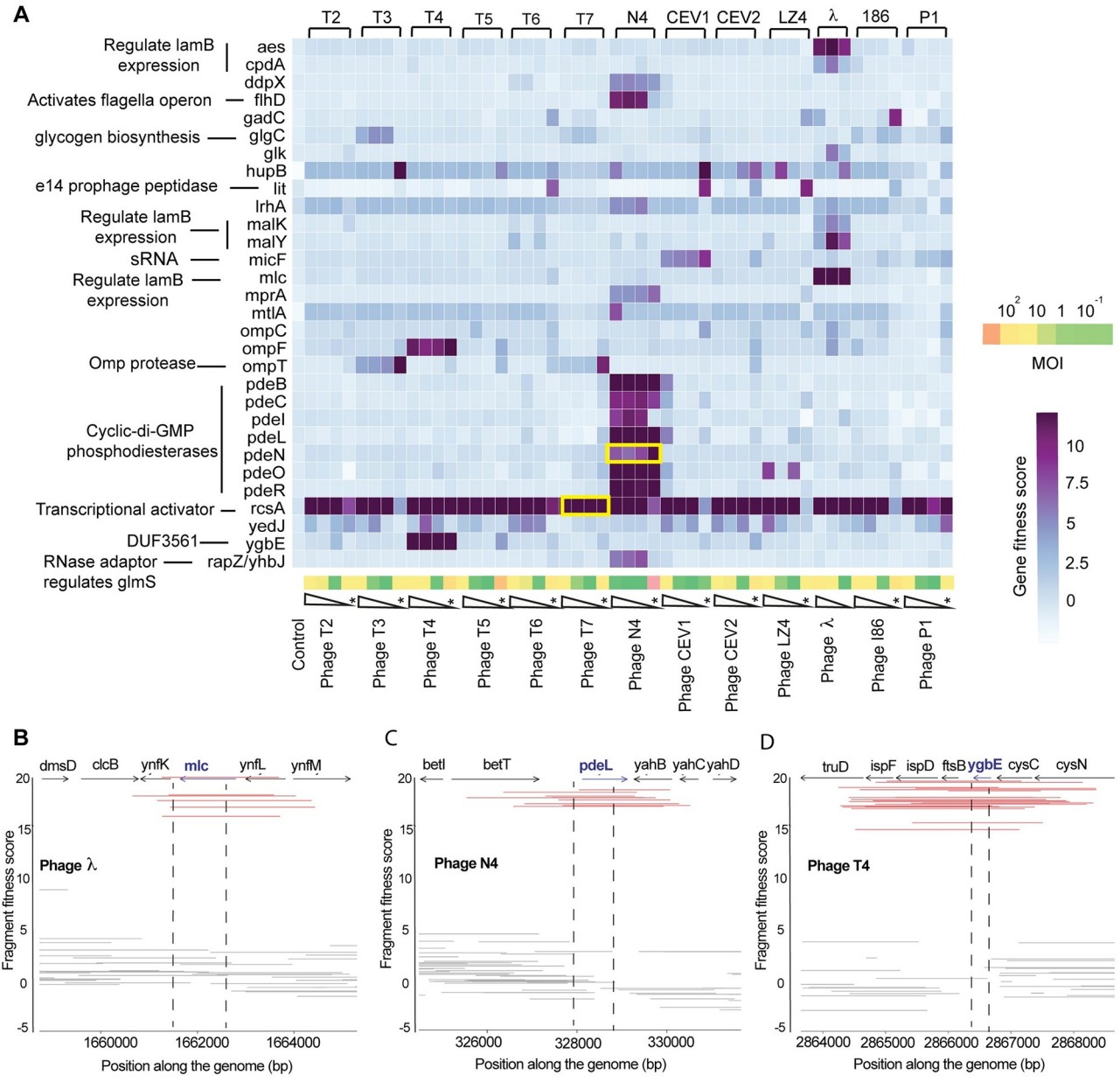

**Fig 4. Dub-seq screening data for 13 dsDNA phages.** (A) Heatmap of Dub-seq data for 13 dsDNA phages at different MOI and 30 genes with large fitness benefits. Overexpression or higher dosage of these genes interferes with the phage infectivity cycle and impart fitness benefits to the host. Only genes with high-confidence effects and gene fitness score of ≥4 in at least one phage assay are shown. Yellow boxes highlight genes that are known to show resistance when overexpressed. The pooled fitness assays performed on solid plate agar are marked with stars. The underlying data for this figure can be found in S1 Data. (B to D) Dub-seq viewer plots for high-scoring *mlc*- (B), *pdeL*- (C), and *ygbE*-containing (D) fragments in the presence of λ, N4, and T4 phages, respectively. Red lines represent fragments covering highlighted genes completely (start to stop codon), whereas gray-colored fragments either cover the highlighted gene partially or do not cover the highlighted gene completely. Additional Dub-seq viewer plots are provided in S2–S4 Figs. dsDNA, double-stranded DNA; Dub-seq, dual-barcoded shotgun expression library sequencing; MOI, multiplicity of infection.

important in the regulation of phage receptor expression and may lead to a phage-resistant strain. To confirm whether our method captures such host factors that control the expression of a phage receptor, we looked for known regulators (Fig 4). Acetylesterase Aes, transcription repressor Mlc, and *mal* regulon repressor MalY are known to reduce the expression of *mal* regulon activator *malT* or prevent MalT's activation of the λ phage receptor *lamB* [141–146].

Indeed, our Dub-seq screen identified *mlc*, *malY*, and *aes* as the top-scoring genes in the presence of λ phage, confirming that Dub-seq can identify host factors involved in regulating the expression of a phage receptor (Fig 4A and 4B). We also found that overexpression of a gene encoding glucokinase (*glk*) and cyclic-AMP (cAMP) phosphodiesterase (*cpdA*) showed enhanced fitness in the presence of λ phage (S2 Fig). Glk has been proposed to inhibit *mal* regulon activator *malT* [147], whereas CpdA hydrolyzes cAMP and negatively impacts cAMP-CRP–regulated gene expression of *lamB* [9,148,149].

One of the top candidates that broadly enhanced host fitness in the presence of diverse phages is transcriptional activator RcsA (that increases colanic acid production by inducing capsule synthesis gene cluster [106]). Genomic fragments with *rcsA* showed the highest gene score of +12 to +16 in most experiments (47/51 assays shown in Fig 4A). In addition, we identified growth advantages with dozens of genes when overexpressed in the presence of specific phages. For example, we found genomic fragments encoding *pdeO (dosP)*, *pdeR (gmr)*, *pdeN (rtn)*, *pdeL (yahA)*, *pdeC (yjcC)*, *pdeB (ylaB)*, *pdeI (yliE)*, *ddpX*, *flhD*, and *yhbJ/rapZ* all confer resistance to N4 phage (Fig 4A and 4C, S4 Fig). Except for *ddpX*, *flhD*, and *yhbJ/rapZ*, which encode D-Ala-D-Ala dipeptidase involved in peptidoglycan biosynthesis, flagellar transcriptional regulator, and an RNase adaptor protein, respectively, all others encode cyclic di-GMP (c-di-GMP) phosphodiesterases (PDEs). The PDEs are a highly conserved group of proteins in bacteria that catalyze the degradation of c-di-GMP, a key secondary signaling molecule involved in biofilm formation, motility, virulence, and other cellular processes [150–152]. The fitness benefit of increased dosage of PDEs is presumably mediated by their degradation of c-di-GMP. We infer that high levels of c-di-GMP are required for phage N4 infection, although the mechanism is unclear. Incidentally, increased dosage of *pdeN* (*rtn)* is known to confer resistance to N4 phage, albeit with unknown mechanism [153,154]. In addition to the N4 phage hits, we found that overexpression of *ygbE* and *ompF* showed high fitness scores in the presence of T4 phages, overexpression of small RNA *micF* showed high fitness scores for CEV1 phage, and overexpression of *ompT* gives high fitness scores in the presence of T3 and T7 phages (Fig 4).

Following earlier work on phage T7 [51], this is the first global survey of how host gene overexpression/dosage impacts resistance to diverse phages belonging to three families. Although we do not understand all of the mechanisms leading to phage resistance, a few of these hits are consistent with the known biology of phage receptors. For example, expression of outer membrane porins *ompC* and *ompF* are regulated reciprocally by *ompR*, and increased *ompF* level reduces expression of the T4 phage receptor *ompC* [155–157]. Similarly, increased expression of the sRNA *micF* causing resistance to phage CEV1 (S2 Fig) is consistent with a report that elevated levels of *micF* specifically down-regulates *ompF* (CEV1 receptor) [158,159]. As *micF* is encoded within the intergenic region of *rcsD* (activator of Rcs pathway and colanic acid biosynthesis) and *ompC* and also contains OmpR operator sites, the resistance-causing Dub-seq fragments containing *micF* could be acting via a combination of effects that cannot be resolved in our screen. Finally, overexpression of the *lit* gene within the defective prophage element e14, shows high fitness in the presence of T6, CEV1, CEV2, 186, λ cI857, and LZ4 phages, but only on plate-based assays (Fig 4, S3 and S4 Figs). Overexpression of *lit* is known to interfere with the T4 phage growth [160], though we did not observe high fitness scores in the presence of T4 phage.

In summary, we identified 129 multicopy suppressors of phage infection that encode diverse functions, and our results indicate that enhanced host fitness (phage resistance) can occur via diverse cellular mechanisms. In the following section, we describe follow-up experiments with 13 of these top-scoring hits from the Dub-seq screen to validate their role in phage resistance.

## Experimental validation of LOF and GOF screen hits

To validate the phage resistance phenotypes observed in our LOF screens, we sourced 6 individual deletion strains *lpcA*, *galU*, *ompF*, *dgcJ*, *ygaH*, and *dsrB* from *E. coli* K-12 BW25113 Keio library [105] and disrupted *igaA* in the *E. coli* K-12 BW25113 strain (S1 Fig). We then determined the efficiency of plating (EOP) for eight phages (Fig 5). In addition, we performed gene complementation experiments using *E. coli* K-12 ASKA plasmids [161] to check if the plating defect can be restored (Fig 5, S5 Fig). To validate the hits identified in our Dub-seq GOF screens, we moved 12 individual plasmids expressing *rcsA*, *ygbE*, *yedJ*, *flhD*, *mtlA*, *pdeB*, *pdeC*, *pdeI*, *pdeL*, *pdeN*, *pdeO*, and *pdeR* into *E. coli* K-12 and tested the EOP of 11 phages (total 36 phage-gene combinations) (Fig 5, Methods). We present these results below.

**CEV1 phage requires both OmpF and LPS for K-12 infection.** *ompF*, *lpcA/gmhA*, and additional genes involved in LPS biosynthesis and transport showed the highest fitness scores in the presence of CEV1 phage in the RB-TnSeq and CRISPRi data (Figs 2A and 3A). In agreement with our LOF screen data, we observed severely reduced EOP of CEV1 on *ompF* and *lpcA* deletion strains compared to the control BW25113 strain (Fig 5A, S5 Fig). These plating efficiency defects could be reverted when the respective deleted genes were expressed in *trans* (S5 Fig). These results indicate that CEV1 infection proceeds by recognizing both OmpF and LPS core, and loss of either *ompF* or LPS disruption leads to a resistance phenotype.

**Overexpression of colanic acid biosynthesis pathway reduces sensitivity to diverse phages.** The Rcs signaling pathway is one of the well-studied signaling pathways in bacteria and is known to regulate a large number of genes involved in diverse functions, including synthesis of colanic acid and biofilm formation in response to perturbations in the cell envelope [106]. Down-regulation of *igaA*, *a* negative regulator of the Rcs signaling pathway, causes resistance to infection by T4, λ, and 186 phages [52] as well as resistance to T7 phage [51]. We found that activation of the Rcs pathway (by either disruption in *igaA* [*yrfF*] or overexpression of *rcsA*) confers resistance to all of the phage we studied. We also observed a high fitness score for sgRNA targeting *dsrB* (a gene of unknown function located downstream of *rcsA*) in CRISPRi screens in the presence of N4 phage (Fig 3A). In agreement with our Dub-seq screen data, we observed the EOP of T2, T3, T4, T5, T6, 186, λ, CEV1, CEV2, LZ4, and N4 phages is compromised when *rcsA* is overexpressed (Fig 5B). Similarly, a *dsrB* deletion strain showed severe N4 phage plating defects, confirming the high fitness scores in our CRISPRi screen (Fig 5A).

Despite *igaA*'s reported essentiality [162,163], our RB-TnSeq library contained 9 disruptions in *igaA*'s cytoplasmic domain, which also overlapped with sgRNA target sites in our CRISPRi screen (Figs 3A and 5A and S1 Fig). To validate that this domain is indeed dispensable for strain viability and also important in phage resistance, we successfully reconstructed the *igaA* insertion mutant (Methods). This mutant strain displayed a mucoidy phenotype indicative of increased activation of colanic acid biosynthesis via the Rcs signaling pathway [106,164]. We observed defective plaque morphologies with T3, T5, T6, T7, P2, 186, CEV1, and N4 phages on the *igaA* insertion mutant, indicating inefficient infection and the plating defect could be reversed by supplying the wild-type *igaA* gene in *trans* (Fig 5A, S5 Fig). To gain further insight into the phage resistance mechanism, we performed RNA sequencing (RNA-seq) analysis on the *igaA* disruption mutant (Methods). We found that multiple components of the Rcs pathway (including *rcsA* itself) were up-regulated, with 24 genes from the capsular biosynthesis-related operons *wca* and *yjb* significantly up-regulated (log2FC > 2, adjusted *p*-value < 0.001) (Fig 6A and 6B, S6 and S7 Tables). These results indicate that the *igaA* disruption mutant uncovered in this work activates colanic acid biosynthesis, leads to a mucoidy phenotype, and may be interfering with phage infection by blocking phage receptor accessibility. The IgaA residues 18–164 we mutate in this work overlap with the N-terminal

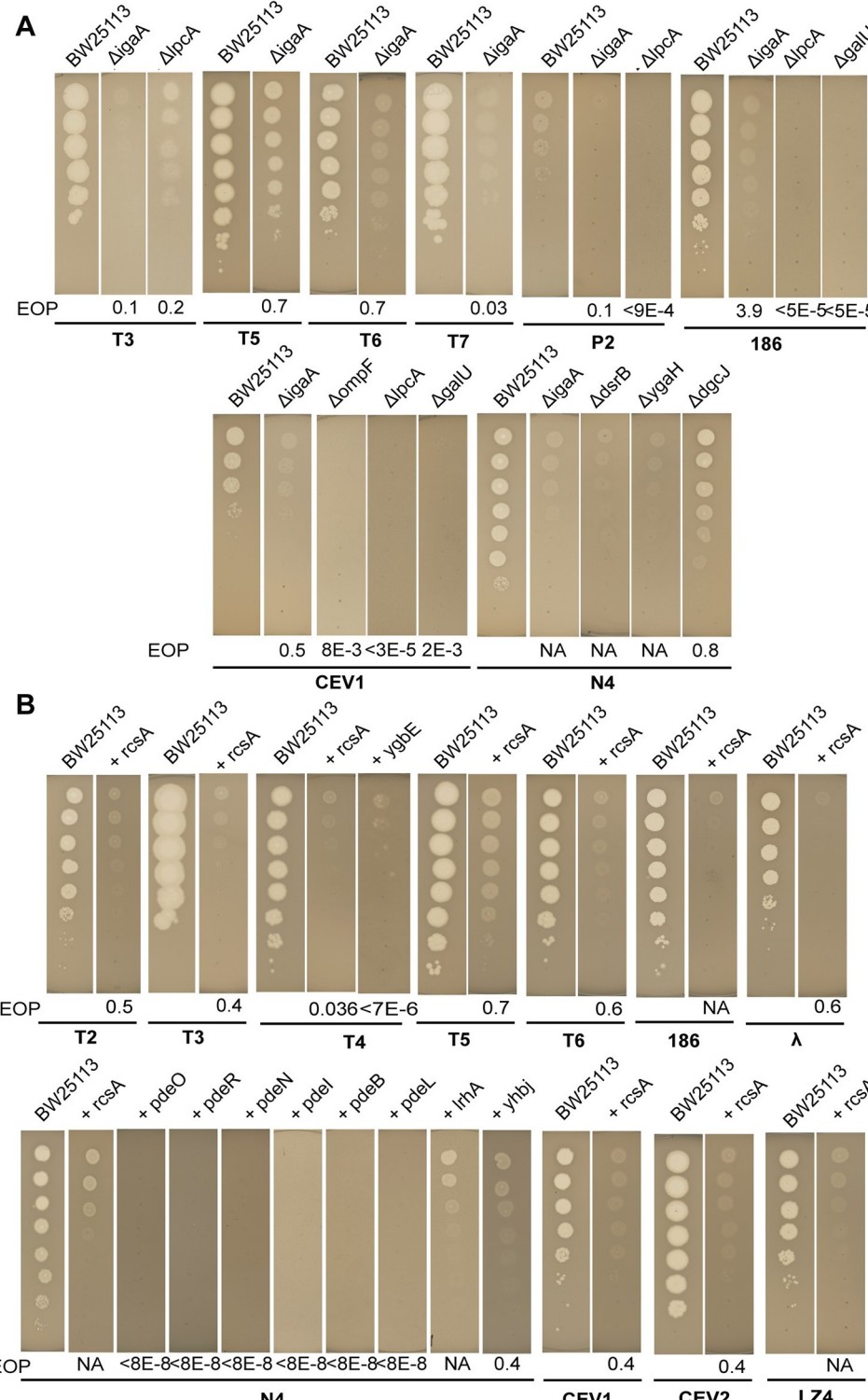

**Fig 5. Experimental validations of top-scoring gene hits in LOF and GOF screens.** (A) EOP experiments for LOF screen hits using Keio [105,161] library strains. Gene complementation data is presented in S2 Fig. (B) EOP experiments for GOF screen hits with ASKA plasmid library [161] expressing genes (shown as +gene names) in the presence of different phages. We used no IPTG or 0.1 mM IPTG for inducing expression of genes from ASKA plasmid. We used the BW25113 strain with an empty vector for estimating EOP. The plaque morphology or EOP of T3, T7, P2, and 186 phages on *lpcA* deletion strain indicated inefficient infection. The plating defect was restored to normal when

mutants were complemented with a plasmid expressing the respective deleted genes indicating LPS core as the receptor for these phages (S5 Fig). EOP, efficiency of plating; GOF, gain-of-function; LOF, loss-of-function; LPS, lipopolysaccharide.

cytoplasmic domain of IgaA that inhibits Rcs signaling in the absence of stress [165]. Recent studies have highlighted that the activation of the Rcs pathway and capsular biosynthesis are essential to survive in diverse environmental contexts, membrane damage, and stress-inducing conditions, including those by antibiotic treatment [106,166]. A number of earlier studies have also highlighted the generation of the mucoid phenotype that probably provides a fitness advantage by blocking phage adsorption [16,167–170]. Our results suggest this might be a generalized mechanism that provides cross-resistance to diverse phages.

**Overexpression of YgbE confers resistance to phage T4 and down-regulates OmpC.** YgbE is a DUF3561-containing inner membrane protein with no known function [171]. In

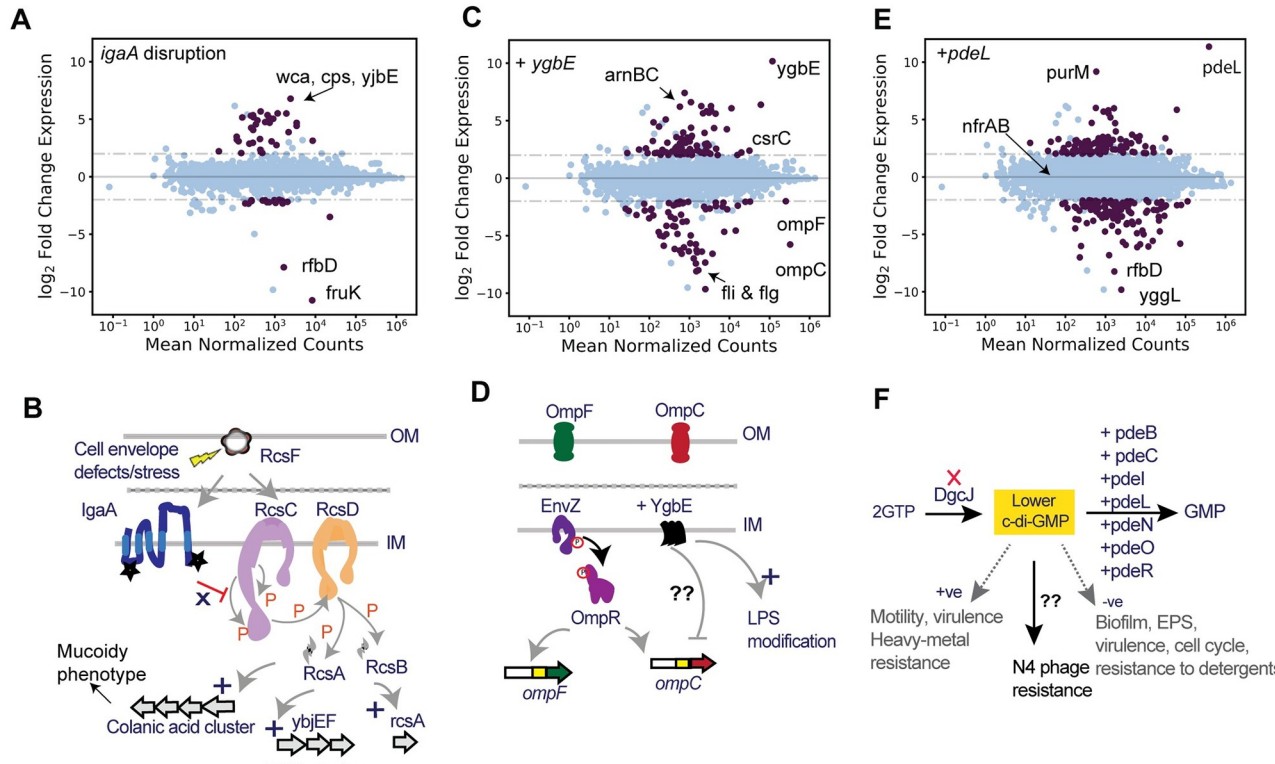

**Fig 6. RNA-seq analysis to gain insights into phage resistance mechanisms.** (A) Up-regulation of EPS biosynthesis genes observed in *igaA* disruption mutant relative to wt (*N* = 3). (B) Schematic of the Rcs phosphorelay pathway. Mutations in *igaA* (shown as stars) identified in this work activate Rcs signaling pathway and induce *rcsA* expression, colonic acid, and EPS biosynthesis pathway. Disruption in *igaA* or overexpression of *rcsA* show a mucoidy phenotype and broad resistance to different phages. (C) Down-regulation of *ompC* transcript and up-regulation of *arnBCA* operon observed during *ygbE* overexpression relative to wt (*N* = 3). (D) Schematic of *ompF* and *ompC* expression regulation via EnvZ-OmpR and YgbE. Overexpression of *ygbE* down-regulates *ompC* expression (via unknown mechanism) and up-regulates genes involved in lipid A modification, probably the reason for resistance to phage T4. (E) RNA-seq data of *pdeL* overexpression showed no down-regulation of N4 phage receptor genes (*nfrA* and *nfrB*) and no up-regulation of genes involved in EPS or biofilm. (F) Schematic of c-di-GMP pathway with *dgcJ* deletion or overexpression of one of 7 PDEs encoding genes (representing decreased c-di-GMP levels) show a high fitness score in the presence of N4 phage via an unknown mechanism that is independent of expression of N4 phage receptor and genes involved EPS biosynthesis. In (A), (C), and (E) plots, purple filled data points are adjusted *p*-value < 0.001 and abs(log2FC) > 2. Blue filled is nonsignificant data points. The dashed lines are effect size thresholds of greater than 4-fold. The underlying data for this figure can be found in S1 Data. c-di-GMP, cyclic di-GMP; EPS, exopolysaccharide; IM, inner membrane; LPS, lipopolysaccharide; OM, outer membrane; PDE, phosphodiesterase; RNA-seq, RNA sequencing; wt, wild type.

Dub-seq screens, *ygbE* shows highest fitness scores in the presence of T4 phage and our EOP data confirm that T4 phage shows strong plating defects when *ygbE* is overexpressed (Fig 5B, EOP of 7E-6). To gain insight into the mechanism of phage resistance, we performed RNA-seq analysis on a *ygbE* overexpression strain. Differential expression analysis of this strain revealed strong down-regulation of *ompC* (log2FC = −5.7, adjusted *p*-value ≪ 0.001), the primary T4 phage receptor (Fig 6C and 6D). In addition, we also noticed a strong down-regulation of 26 genes (log2FC = −4.5 adjusted *p*-value ≪ 0.001) related to flagella structure including RNA polymerase sigma 28 (sigma F) factor, and strong up-regulation (log2FC = 4.5 adjusted *p*-value ≪ 0.001) of *arnBCADT* operon involved lipid A modification (Fig 6C and 6D, S6 and S7 Tables). The down-regulation of *ompC* and up-regulation of LPS modification genes is in agreement with the observed phage resistance phenotype of *ygbE* (Figs 4A, 4D and 5B), but the mechanism of *ompC* down-regulation in the *ygbE* overexpression strain remains to be determined.

**c-di-GMP is required for infection by phage N4.** c-di-GMP is a key bacterial secondary signaling molecule involved in regulation of diverse cellular functions. The top eight hits in our LOF and GOF screens for N4 phage resistance included enzymes that catalyze synthesis and degradation of c-di-GMP. Diguanylate cyclase J (encoded by *dgcJ*) is involved in the biosynthesis of c-di-GMP and is one of the top scorers in both RB-TnSeq and CRISPRi LOF screens, whereas the seven c-di-GMP-specific PDEs that are involved in degradation of c-di-GMP showed the highest fitness scores in the Dub-seq screen. Though the signaling network of c-di-GMP is complex, deletion of DGCs or overexpression of PDEs is known to reduce c-di-GMP levels, inhibit biofilm formation, reduce biosynthesis of curli, and increase motility [150–152,172]. The high fitness scores for *dgcJ* in RB-TnSeq and CRISPRi screens is intriguing considering it is one of the 12 DGCs encoded on *E. coli* K-12 genome [173,174], and none of the other DGCs show phenotypes in our screens. Similarly, *E. coli* K-12 genome codes for 13 PDEs in total [173,174], and we find six of these PDEs show a phage resistance phenotype when overexpressed. Our EOP estimations with N4 phage showed severe plating defect (EOP < 8E-8) on *pdeO (dosP)*, *pdeR (gmr)*, *pdeN (rtn)*, *pdeL (yahA)*, *pdeB (ylaB)*, and *pdeI (yliE)* overexpressing strains and minor plating defect on *dgcJ* mutant (EOP of 0.8) (Fig 5A and 5B). The plating defect of N4 phage on *dgcJ*::*kan* could be reverted back when the *dgcJ* was provided in *trans* (S5 Fig).

To gain insight into how the overexpression of specific genes might affect phage infection pathways, we performed differential RNA-seq experiments (standard growth conditions, in the absence of phage) on five c-di-GMP PDE (*pdeL*, *pdeB*, *pdeC*, *pdeN*, and *pdeO*) overexpressing strains, each of which shows resistance to N4 phage (Methods). RNA-seq experiments on the *pdeL* overexpression strain revealed large changes in the *E. coli* transcriptome relative to the wild-type BW25113 strain, with 103 genes significantly up-regulated (log2FC > 2, q < 0.001) and 109 genes significantly down-regulated (log2FC < −2, q < 0.001) (Fig 6E and 6F, S6 and S7 Tables). N4 phage receptor genes *nfrA* and *nfrB* were not differentially expressed. Similar to *pdeL* overexpression, overexpression of the other four PDEs also did not show substantially different expression of *nfrA* and *nfrB*, suggesting that N4 resistance phenotype in these instances does not appear to be via transcriptional regulation of the N4 phage receptor genes (S6 and S7 Tables). Further studies are underway to gain more insights on how c-di-GMP levels influence N4 phage infection cycle.

**Validation of other top-scoring genes displaying N4 phage resistance phenotype.** *ygaH* is one of the four top-scoring RB-TnSeq candidates in the presence of N4 phage (Fig 2A). YgaH is predicted to be a L-valine exporter subunit [171,175] and had not been previously associated with N4 phage infection. In agreement with its strong fitness scores in RB-TnSeq data, our EOP estimations on *ygaH* deletion strain confirmed severe plating defects on N4

phage (Fig 5A). Although the role of YgaH in N4 phage infection pathway is unclear, Dub-seq fragments encoding *mprA* (a negative regulator of multidrug efflux pump genes), a gene downstream of *ygaH*, also shows an N4 phage resistance phenotype (Fig 4, S3 Fig), further demonstrating the importance of this region in N4 phage infection.

Among other top hits in the GOF Dub-seq screen, we observed defective plating of N4 phage on *lrhA* and *rapZ* (*yhbJ*) overexpression strains (Fig 5B, S3 Fig). *lrhA* codes for a transcription regulator of genes involved in the synthesis of type 1 fimbriae, motility, and chemotaxis [176] and has not been previously linked to N4 phage infection. *rapZ* codes for an RNase adaptor protein that negatively regulates the expression level of glucosamine-6-phosphate synthase (GlmS) [176,177]. GlmS catalyzes the first rate-limiting step in the amino sugar pathway supplying precursors for assembly of the cell wall and the outer membrane. Though the role of the essential gene *glmS* in N4 phage infection is unclear, the N4 phage resistance phenotype shown by *rapZ* multicopy expression (which down-regulates *glmS* expression) agrees with our CRISPRi screen data wherein knockdown of both *glmS* and *glmU* within *glmUS* operon show strong fitness in the presence of N4 phage (Fig 3C). These results illustrate the power of using these combination technologies for studying phage infection.

Finally, candidates that showed high fitness scores in our Dub-seq screen but failed to demonstrate strong phage plating defects include strains overexpressing *flhD* or *mtlA* in the presence of N4 phages and *yedJ* in the presence of multiple phages (Fig 4A, S6 Fig). We speculate other entities such as sRNA coding regions or transcription factor binding sites on Dub-seq fragments encoding these gene loci may be essential for phage resistance phenotype (S6 and S7 Figs).

## Extending genome-wide screens to *E. coli* BL21 strain

To compare the phage resistance phenotypes we observed in *E. coli* K-12 to a closely related host, we chose *E. coli* BL21 strain as our alternate model system. The genomes of BL21 and K-12 strains have 99% bp identity over 92% of their genomes, interrupted by deletions or disruptions (by mobile elements) [88–90]. For example, in comparison to K-12, BL21 has a disruption or deletions in the colanic acid pathway, biofilm formation, flagella formation (a 21-gene cluster including flagella-specific *fliA* sigma factor), genes involved in Rcs signaling pathway (*rcsA*, *rcsB*, *rcsC*), the LPS core gene cluster (BL21 forms truncated LPS core with only two hexose units compared to normal five hexose units in K-12, Fig 2B), and is deficient in Lon and OmpT proteases, which are important components of the protein homeostasis network [88–90]. In addition, BL21 also lacks *ompC* and *nfrAB* genes that code for T4 phage and N4 phage receptors in *E. coli* K-12, respectively [90,178,179].

To investigate BL21 genes essential for phage growth, we constructed an RB-TnSeq library made up of approximately 97,000 mutants (Methods). For fitness assays, we used the same set of phages that were assayed with K-12 library except phages N4 and 186. Both N4 and 186 phages do not infect BL21 because of a lack of N4 phage receptors (NfrA and NfrB) and truncated LPS that probably limits 186 binding. We performed 53 pooled fitness experiments using the BL21 RB-TnSeq library in both liquid and solid plate assay format at varying MOI and nine no-phage control experiments. In total, we identified 115 high-scoring hits, made up of 50 unique phage-gene combinations and representing 32 unique genes (S8 Table). All 12 phages have at least one high-confidence hit. Largely, the BL21 LOF data are in agreement with our K-12 results for T3, T5, T6, T7, CEV2, LZ4, λ, P1, and P2 phages, especially the high-scoring genes that either code the phage receptor or its regulators (Fig 7).

The key differences between BL21 and K-12 LOF fitness data are the host factors important in T2, T4, and CEV1 phage infection. For example, *ompC* and its regulator *EnvZ/OmpR* two-

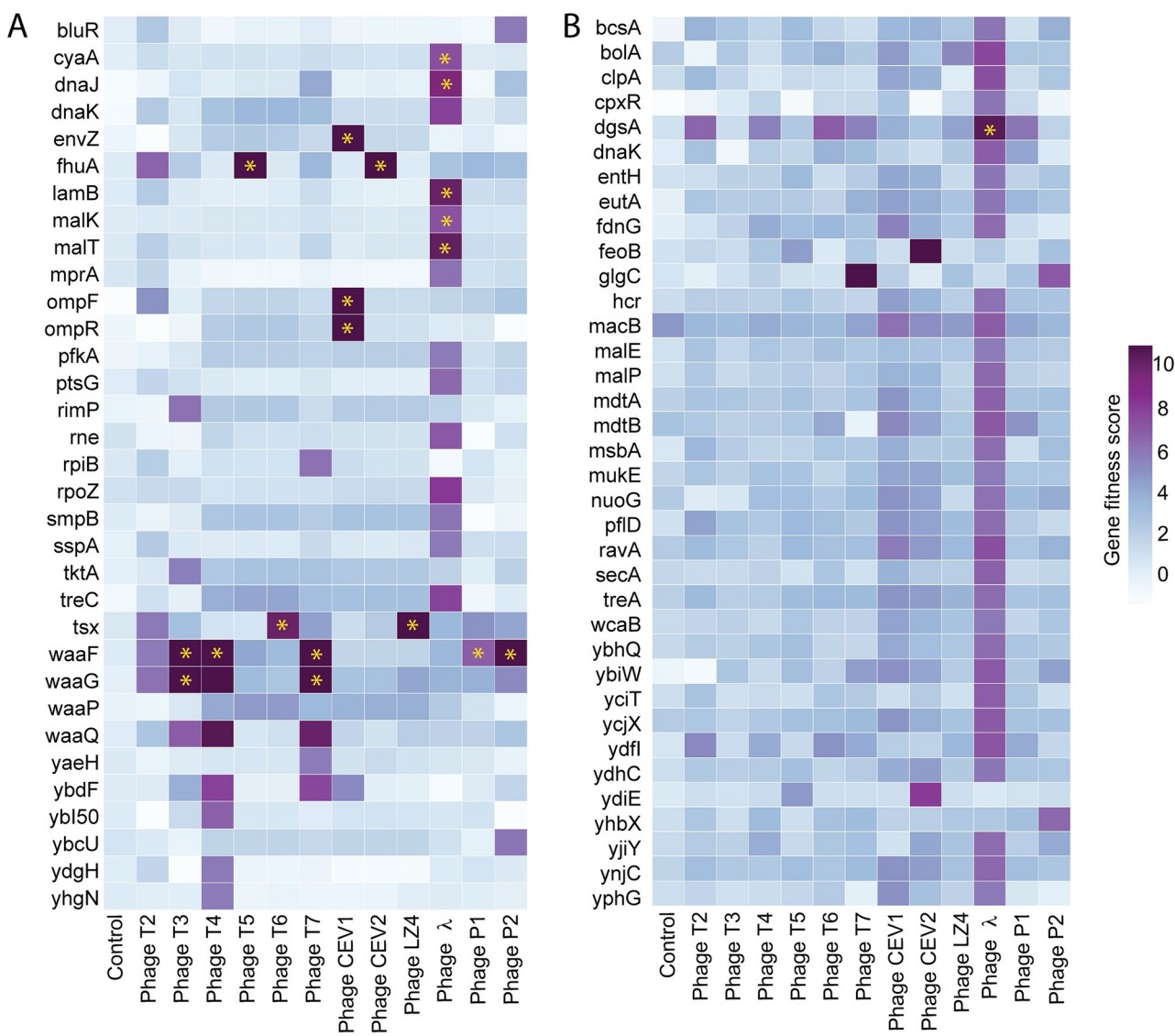

**Fig 7. Genome-wide screens in *E. coli* BL21 strain.** (A) Heatmap of BL21 LOF RB-TnSeq data for 12 dsDNA phages at a single MOI, and selected genes with high-confidence fitness benefits are shown. (B) Heatmap of GOF BL21 Dub-seq data for 12 dsDNA phages with high-confidence fitness benefit. Fitness scores of ≥4 in at least one phage assay are shown. These assays were performed in planktonic culture. Yellow stars indicate these data points are in agreement with RB-TnSeq, CRISPRi, and Dub-seq data for *E. coli* K-12. The underlying data for this figure can be found in S1 Data. CRISPRi, CRISPR interference; dsDNA, double-stranded DNA; Dub-seq, dual-barcoded shotgun expression library sequencing; GOF, gain-of-function; LOF, loss-of-function; MOI, multiplicity of infection; RB-TnSeq, random barcode transposon site sequencing.

component system and genes involved LPS core biosynthesis and showed high fitness scores in the presence of T4 phage in K-12 LOF screens, whereas only genes involved in the LPS core biosynthesis (*waaG*, *waaF*, and *waaQ*) were important in BL21 screens (Fig 7). Lack of *ompC* in the BL21 strain probably alleviates the need for the EnvZ/OmpR two-component system for T4 infection. The absolute requirement for LPS R-core structure for T4 phage growth on BL21 is in agreement with the earlier reports [91,92,179–181]. Similarly, CEV1 phage, which showed a strict requirement for OmpF and full-length LPS in K-12 screens (Fig 2), seems to require only OmpF in the BL21 infection cycle (Fig 7). This suggests that CEV1 phages can tolerate truncated LPS of BL21 but not that of K-12 (Fig 2). Because of the OmpF requirement, CEV1

growth on BL21 also showed strict dependence on the EnvZ/OmpR two-component system, a key regulator of *ompF* expression. Finally, in distinction to K-12 data, our BL21 RB-TnSeq data indicate that T2 phage probably binds preferentially to truncated LPS R-core in BL21 (*waaF* and *waaG* showed high fitness in our screen). Furthermore, this effect seems to be independent of FadL. We confirmed this observation by measuring the EOP on a BL21 *fadL* deletion strain. The absence of a FadL requirement for T2 growth on BL21 is intriguing considering its 100% nucleotide identity with K-12 *fadL*. It is possible that either the conformational integrity of FadL is compromised in the absence of full-length LPS or that T2 may be recognizing more than one outer membrane protein [95,182] (S1 Text). These results are consistent with earlier observations on the difficulty in isolating T2-resistant mutants in *E. coli* B cells [15].

Finally, to investigate whether increased gene copy number of host factors interferes with phage growth in BL21, we constructed a BL21 Dub-seq library. This library is made up of randomly sheared BL21 genomic DNA cloned into a dual-barcoded vector and is made up of a total 65,788 unique 3-kb fragments (Methods, S8 Fig). We then screened the BL21 Dub-seq library in a variant of BL21 as the host (Methods). From 24 pooled fitness experiments in planktonic cultures and on solid media in the presence of 12 phages, we identified 39 high-scoring candidates (fitness score ≥ 4, FDR of 0.74, Methods). Other than a few top-scoring candidates in the presence of λ phage, the BL21 dataset was considerably different from K-12 dataset (S9 Table).

Some of the top-scoring hits in BL21 Dub-seq screen showed broad resistance to many different phages whereas some were phage specific. For example, BL21 Dub-seq fragments with *mlc* (*dgsA*) gave higher fitness in the presence of T2, T4, T6, T7, λ, and P1 phages. This resistance phenotype of Mlc in the presence of λ phage is consistent with its known regulatory role. Mlc, a global regulator of carbohydrate metabolism, is known to negatively regulate the maltose regulon and mannose permease system [183] (via regulating the *lamB* expression activator MalT), both of which are known to play a crucial role in phage λ DNA penetration and infection [184–187]. Another top-scoring candidate *glgC* showed higher fitness in the presence of T7 and P2 phages. Overexpression of *glgC* (which encodes Glucose-1-phosphate adenylyltransferase) is known to increase glycogen accumulation [188,189] and titrate out the global carbon storage regulator CsrA [190]. We speculate that the interaction of GlgC and CsrA probably impacts biofilm formation [130,191–193] and leads to alternations in the LPS profile [194,195], leading to phage resistance phenotype. In agreement to our results for *E. coli* K-12 BW25113 and MG1655 fitness assays, we did not observe enrichment of host factors that may enhance the rate of lysogeny of 186 and P2 phages.

Among the candidates that show phage specific resistance phenotype, we observed that overexpression of *argG* showed a fitness score of 3.7 in the presence of T4 phages (S9 Table). Argininosuccinate synthetase enzyme (encoded by *argG*) catalyzes the penultimate step of arginine biosynthesis and has not been associated with phage resistance phenotype before. However, early studies have indicated the inactivating effect of arginine on T4 phages [196]. Finally, Dub-seq fragments encoding ferrous iron uptake system (FeoB) and putative heme trafficking protein (YdiE) yield strong fitness in the presence of T5 phage and T5-like CEV2 phage (Fig 7). It is known that increased uptake of ferrous iron increases Fur-ferrous iron occupancy and Fur-mediated repression of T5 phage receptor *fhuA* [197–201]. Most of these top-scoring candidates were missing in the K-12 dataset, probably because of strong selection for *rcsA* overexpressing strains in all of our K-12 Dub-seq experiments. This highlights the importance of studying how even closely related hosts can have nuanced interactions with the same bacteriophage.

## Summary and conclusions

We applied unbiased high-throughput LOF and GOF screening methods to two different *E. coli* strains to map the landscape of genetic determinants important in host-phage interactions. We demonstrate how these methods can rapidly identify phage receptors and both novel and previously described non-phage-receptor-related host factors involved in resistance across a wide panel of dsDNA phage types. By using LOF RB-TnSeq and CRISPRi methods, we extensively map both nonessential and essential host genes along with nongenic regions such as promoter and transcription factor binding sites implicated in phage infection and resistance. The Dub-seq methodology uncovers dozens of multicopy suppressors that encode diverse functions and point to a myriad of ways how host gene dosage can influence phage resistance. This global survey of host factors that play an important role in phage growth across two widely studied *E. coli* strains provides a detailed view of cross-resistance patterns for diverse phages and will be a rich dataset for deeper biological insights and machine learning approaches.

Our data from both LOF and GOF screens show consistency across a range of MOI and also suggest that different assay formats (solid and liquid) allow increased discovery of diverse phage resistance mechanisms. These assay platforms could be expanded to phage-banks made up of hundreds of phages at just one MOI and may be sufficient to rapidly discover the phage receptors in the target host. In addition, extending the screening methods to closely related but different strains would be highly valuable. For example, we used two laboratory *E. coli* strains that have rough LPS architecture where core oligosaccharide is the terminal part of the LPS. However, it is known that the genetic and structural diversity of LPS and the repeat structure O-polysaccharide attached to LPS (to form smooth LPS) is very large in pathogenic and environmental isolates of Enterobacteriaceae and may impact phage infectivity [48,58,202–205]. Although we used nonpathogenic *E. coli* in this work, we did study two phages (CEV1 and CEV2) that infect pathogenic O157:H7 strains and identified key host factors important for growth of these phages. Our study provides an opportunity to compare these host factor requirements for CEV1 and CEV2 with their structurally similar non-pathogen-associated phages T4 and T5, respectively. The genetic screens presented in this work and future extension to diverse *E. coli* serogroups may aid in filling the knowledge gap on phage interaction with different antigens and its impact on phage infectivity and resistance.

Our results highlight that phage infectivity depends on the host cellular physiology and, in particular, membrane characteristics of the host imparted by LPS and outer membrane protein biogenesis pathways [113,114,206]. For instance, our results indicate that the disruptions in LPS (for example, deletions or down-regulation of *waa* genes Figs 2A, 3A and 7A), LPS transport pathways (for example, down-regulation of *lptABC*, Fig 3B), and signal resembling membrane stress (for example, disruption in *igaA* in Figs 2A and 3B; overexpression of *rcsA*, *ompF*, *micF*, in Fig 4A) can influence phage infectivity (additional discussion in S1 Text). We also observed strain-specific similarities and differences in the phage infectivity patterns, as well as environmentally context dependent differences. These results indicate the importance of studying phage resistance across a diverse set of biotic and abiotic conditions and how they impact the cross-resistance and cross-sensitivity in closely and distantly related organisms. For example, gaining insights into conditions under which a bacterium would actively overexpress the colanic acid biosynthesis pathway may help us to develop therapies to overcome the generalized phage defense mechanism.

The strong positive fitness scores we observe due to the selection pressure during these pooled fitness experiments are both a strength and limitation of our methods. The strength of these screens is that we can rapidly identify host factors crucial in phage infectivity cycle because they display a very strong fitness score when disrupted (for example, a phage receptor

and its regulators in LOF screen). This strong positive selection, however, limits the detection of intermediate phage resistance routes (for example, effect of *trxA* and *cmk* deletion on T7 phage growth), whose disruption may lead to strains with relatively poor or neutral fitness and will be swept from the population. We addressed this limitation by using solid agar plate assays, where direct competition among resistant types may be reduced. Future fitness assays can be extended to droplet microfluidics platforms [207] to uncover intermediate fitness phenotypes and retain the high-throughput scalability of these approaches. We note here that the relative fitness of these lab-generated strains may be distinct from the likelihood that their cognate mutant will naturally evolve under phage selection pressure [208]. In addition, we also did not identify strong effects for bacterial defense mechanisms (for example, R-M and CRISPR systems). In both *E. coli* K-12 BW25113 and BL21, CRISPR and type I R-M systems are disrupted but other R-M systems such as McrA, McrBC, and Mrr are intact [88,90,105]. Other than strains with high-scoring *mcrB* (encodes subunit of McrBC system, S1 Text, S3 Fig) in the presence of T2 phage, we did not see high fitness scores for other restriction systems in our Dub-seq fitness screens. We speculate that the expression of restriction enzyme subunit has to be at optimum level to counter the phage infectivity cycle and also to compete with high fitness strains (such as strains with higher RcsA expression) in the pooled fitness assays. One way to improve the detection of overexpression gene phenotypes is to use phage mutants that lack anti-R-M systems and nonessential regions. For example, T7 phage mutants lacking gene 0.3, gene 1.2, and region 0.3–1.4 were used earlier to uncover the overexpression phenotype of *hsdR* (constitutes the restriction subunit of EcoKI), *dgt* (dGTPase), and *udk* (encodes for uridine/cytidine kinase) on T7 phage growth, respectively [51,209].

The screening technologies presented in this work are scalable to study phage resistance mechanisms in diverse organisms. Two key considerations in this regard that need to be accounted for are (1) the availability and extendibility of genetic tools to the organisms of interest and (2) the cost and time in creating the RB-TnSeq, CRISPRi, and Dub-seq libraries. Based on our experience, the cost of creation of these libraries falls in the range of $4,000–$12,000 per library, and it takes about 3 mo on average for building fully characterized libraries. The newly developed sequencing technologies and extendibility of genetic tools to nonmodel organisms can further reduce the time and cost of generating the LOF and GOF libraries [70,210,211]. Once the libraries are built, the running cost of these genome-wide assays across 96 conditions cost about $10 per assay and serve as a standardized reagent tool for in-house experiments or for sharing and comparing data across different laboratories. The ease and economics of these genetic screens enable extendibility of phage resistance assays in diverse conditions that simulate the natural environment and may provide valuable insights on host fitness and host-phage coevolutionary dynamics under more ecologically relevant conditions. For example, recent studies highlighted the evidence of subdued phage resistance in the natural environment, probably because of the fitness cost associated with resistance mechanisms [97,208,212–216]. In addition, these methods have the capability to identify fitness costs associated with broadly seen phage resistance phenotypes in a competitive natural environment and thus improve our understanding of microbial ecology in general [13,119,213,214,217–219]. Such systems-level insights will be valuable both in uncovering new mechanisms in host-phage interaction and perhaps in developing different design strategies for targeted microbial community interventions, engineering highly virulent or extended host-range phages and rationally formulated phage cocktails for therapeutic applications [97,212,215,220–233]. Alternatively, identifying phage resistance determinants may also enable engineering of bacterial strains with phage defense systems crucial in a number of bioprocesses such as in the dairy industry [234,235], biocontainment strategies for bioproduction industry [236,237], or facilitation of bacterial vaccine discovery and development [238–240].

There is a clear applied interest in utilizing combinations/cocktails of phages to regulate or eliminate bacterial populations due to the reduced likelihood of evolved multiphage resistance [97,221,225]. However, designing such cocktails relies on a better understanding of cross-resistance among phages [28,61,241,242]. In particular, identification of phages that differ in their receptor use or against which cross-resistance is unlikely to evolve would allow for better design of such therapies [23,37,242]. Moreover, identifying phages that select for resistance that have interrelated phenotypic consequences with, for example, antibiotic sensitivity is a recent advancement in the field that could directly benefit from these screening approaches [33,53,97,243]. By combining fitness datasets for phages and antibiotics or phage-antibiotic combination therapies [244–246], such screens could provide an avenue for performing high-throughput search for genetic trade-offs or "evolutionary traps" [33,53,97,243] that could provide a much-needed solution to overcome the antibiotic-resistance pandemic.

## Methods

### Bacterial strains and growth conditions

The primers and plasmids used in this study are listed in S10 and S11 Tables, respectively. The bacterial strains, phages, and their sources are listed in S12 Table. Phage plaque-forming units/ml and MOIs used in each experiment are listed in the S13 Table. All plasmid manipulations were performed using standard molecular biology techniques [247]. All enzymes were obtained from New England Biolabs (NEB) and oligonucleotides were received from Integrated DNA Technologies (IDT). Unless noted, all strains were grown in LB supplemented with appropriate antibiotics at 37 ˚C in the Multitron shaker. All bacterial strains were stored at −80 ˚C for long-term storage in 15% sterile glycerol (Sigma). The genotypes of *E. coli* strains used in the assays include BW25113 (K-12 lacI+rrnBT14 Δ(araB–D)567 Δ(rhaD–B)568 ΔlacZ4787(::rrnB-3) hsdR514 rph-1), *E. coli* MG1655 (K-12 F–λ–ilvG–rfb-50 rph-1) and *E. coli* BL21 (B F–ompT gal dcm lon hsdSB(rB–mB–) [malB+]K-12(λS)). The genetic and phenotypic differences between these strains are well documented [88–90,105,248]. We note here that all three strains lack an active CRISPR system. Both *E. coli* K-12 BW25113 and BL21 have disruption in the type I R-M system (known as EcoKI in *E. coli* K-12 BW25113 and EcoBI in *E. coli* Bl21), encoded by the *hsdRMS* genes, but have functional McrA, McrBC, and Mrr restriction systems, encoded by *mcrA*, *mcrBC*, and *mrr* [90]. *E coli* MG1655 has functional EcoKI, McrA, McrBC, and Mrr restriction systems.

### Bacteriophages and propagation

The bacteriophages used in this study and their sources are listed in S12 Table. All phages except P2 phage and N4 phage were propagated on *E. coli* BW25113 strain. To propagate P2 phage and N4 phage, we used *E. coli* C and *E. coli* W3350 strains, respectively. We used a host-range mutant of T3 (from our in-house phage stock that can grow on both *E. coli* K-12 and BL21 strains), mutant λ phage (temperature-sensitive mutant allele cI857) [9,249], and a strictly virulent strain of P1 phage (P1*vir*) [250] that favors a lytic phage growth cycle. We followed standard protocols for propagating phages [231]. Phage titer was estimated by spotting 2 μl of a 10-fold serial dilution of each phage in SM buffer (Teknova) on a lawn of *E. coli* BW25113 via top agar overlay method using 0.7% LB agar. SM buffer was supplemented with 10 mM calcium chloride and magnesium sulphate (Sigma). We routinely stored phages as filter-sterilized (0.22 μm) lysates at 4 ˚C.

## Construction of BL21 RB-TnSeq library

We created the *E. coli* BL21-ML4 transposon mutant library by conjugating *E. coli* BL21 strain with *E. coli* WM3064 harboring the pKMW3 mariner transposon vector library (APA752) [64]. We grew *E. coli* BL21 at 30 ˚C to mid-log-phase and combined equal cell numbers of BL21 and donor strain APA752, conjugated them for 5 hr at 30 ˚C on 0.45-m nitrocellulose filters (Millipore) overlaid on LB agar plates containing diaminopimelic acid (DAP) (Sigma). The conjugation mixture was then resuspended in LB and plated on LB agar plates with 50 μg/ml kanamycin to select for mutants. After 1 d of growth at 30 ˚C, we scraped the kanamycin-resistant (kanR) colonies into 25 ml LB, determined the OD600 of the mixture, and diluted the mutant library back to a starting OD600 of 0.2 in 250 ml of LB with 50 μg/ml kanamycin. We grew the diluted mutant library at 30 ˚C for a few doublings to a final OD600 of 1.0, added glycerol to a final volume of 15%, made multiple 1-ml −80 ˚C freezer stocks, and collected cells for genomic DNA extraction. To link random DNA barcodes to transposon insertion sites, we isolated the genomic DNA from cell pellets of the mutant libraries with the DNeasy kit (Qiagen) and followed published protocol to generate Illumina compatible sequencing libraries [64,68]. We then performed single-end sequencing (150 bp) with the HiSeq 2500 system (Illumina). Mapping the transposon insertion locations and the identification of their associated DNA barcodes was performed as described previously [64]. Of 4,195 protein-coding genes in *E. coli* BL21, our BL21 RB-TnSeq library has fitness estimates for 3,083. Twelve independent strains were used to compute fitness for the typical protein-coding gene.

## Construction of BL21 Dub-seq library

To construct *E. coli* BL21 Dub-seq library, we used the dual-barcoded pFAB5526 plasmid library with a kanamycin resistance marker (https://benchling.com/s/seq-1Gkg3lrrSno4EF0Ye11k). The Dub-seq backbone plasmid pFAB5526 is the same in design and genetic composition to the original pFAB5491 Dub-seq plasmid (https://benchling.com/s/seq-39Hoh4d1AResilOUPVJ9) [64,66] except for the kanamycin resistance marker and a mobility gene present on pFAB5526. We mapped the dual barcodes of the pFAB5526 library via the Barcode-Pair sequencing (BPseq) protocol [64,66]. We sequenced BPseq samples on HiSeq 2500 system with 150-bp single-end runs. We then cloned 3 kbp of *E. coli* BL21 genomic fragments between UP and DOWN barcodes by ligating the end-repaired genomic fragments with PmlI restriction digested pFAB5526, electroporating the ligation into *E. coli* DH10B cells (NEB) and selecting the transformants on LB agar plates supplemented with kanamycin (50 μg/ml). We scraped the kanR colonies into 25 ml LB and diluted the transformant mixture to a starting OD600 of 0.2 in 250 ml of LB with 50 μg/ml kanamycin and grew the library at 30 ˚C for few doublings to a final OD600 of 1.0. Finally, we added glycerol to a final volume of 15%, made multiple 1-ml −80 ˚C freezer stocks, and collected cells for plasmid DNA extraction (Qiagen). Next, we mapped the cloned genomic fragment and its pairings with neighboring dual barcodes via Barcode Association with Genome fragment sequencing (BAGseq) [64,66]. We sequenced the BAGseq samples on HiSeq 2500 system with 150-bp single-end runs using the reported sample preparation steps [64,66]. The data processing of BPseq and BAGseq steps was performed using *Dub-seq* python library with default setting (https://github.com/psnovichkov/DubSeq), as detailed earlier [64,66]. We mapped *E. coli* BL21 Dub-seq library to *E. coli* BL21-DE3 genome sequence [88]. BL21 Dub-seq library was then electroporated into BL21DE3C43 strain and the transformants were collected into 25 ml LB. The transformant mixture was then diluted to a starting OD600 of 0.2 in 250 ml of LB with 50 g/ml kanamycin and grew the library at 30 ˚C for a few doubling to a final OD600 of 1.0. Finally, we added glycerol to a final volume of 15%, made multiple 1-ml −80 ˚C freezer stocks. These stocks were

further used for pooled fitness assays as described below. The BL21 Dub-seq library is made up of 65,788 unique barcoded fragments. The average fragment size of the library is 2.5 kb and the majority of fragments covered 2–3 genes in their entirety (S8 Fig). Similar to *E. coli* BW25113 Dub-seq library [64,66], BL21 Dub-seq library covers 85% of genes from start to stop codon by at least five independent genomic fragments, and 97% of all genes are covered by at least one fragment. In total, 132 genes are not covered in their entirety by any Dub-seq fragment.

## Competitive growth experiments with RB-TnSeq library

A single aliquot (1 ml) of a mutant library was thawed, inoculated into 25 ml of medium supplemented with kanamycin (50 μg/ml), and grown to OD600 of 0.6–0.8 at 37 ˚C. After the mutant library recovered and reached mid-log phase, we collected cell pellets as a common reference for BarSeq (termed time-zero or start samples) and used the remaining cells to set up competitive mutant fitness assays in the presence of different phages at different MOI. For performing planktonic culture assays, we diluted the recovered mutant library stock to a starting OD600 of 0.04 in 2X LB media (350 μl) and mixed in equal volume (350 μl) of phages diluted in phage dilution buffer. We also set up control "no-phage" competitive mutant fitness assays wherein we replaced phages with simply the phage dilution buffer. The mutant library experiments were grown in the wells of a 48-well microplate (700 μl per well). We grew the microplates in Tecan Infinite F200 readers with orbital shaking and OD600 readings every 15 min for 8 hr. After the experiment, survivors were collected, pelleted, and stored at −80 ˚C prior to genomic DNA extraction. For solid plate-based assays, we incubated the mixture of phage and diluted mutant library at room temperature for 15 min and then plated the mixture on LB agar supplemented with kanamycin plates and incubated at 37 ˚C overnight. The next day, the resistant colonies were scraped, resuspended in 1 ml LB media, and pelleted. Assay pellets were typically stored at −80 ˚C prior to genomic DNA extraction. The 8-hr assay period was decided based on our preliminary time-course experiment results for a specific phage at different MOIs, wherein we took intermittent samples between 2 and 10 hr, processed as detailed below and found to yield consistent top-scoring hits. In addition, assay samples after 8 hr provided sufficient DNA for the sample processing step.

## Competitive growth experiments with Dub-seq library

Similar to RB-TnSeq competitive fitness assays, a single aliquot of the *E. coli* Dub-seq library (*E. coli* BW25113 library expressed in *E. coli* BW25113 or *E. coli* BL21 Dub-seq library in *E. coli* BL21DE3C43 strain) was thawed, inoculated into 25 ml of LB medium supplemented with appropriate antibiotics (chloramphenicol 30 μg/ml for *E. coli* BW25113 Dub-seq library and kanamycin [50 μg/ml] for *E. coli* BL21 Dub-seq library), and grown to OD600 of 0.6–0.8. At mid-log phase, we collected cell pellets as a common reference for BarSeq (termed start or time-zero samples), and we used the remaining cells to set up competitive fitness assays in the presence of different phages at different MOI. For performing planktonic culture assays, we diluted the recovered Dub-seq library stock to a starting OD600 of 0.04 in 2X LB media and mixed in equal volume (350 μl) with phages diluted in phage dilution buffer. We also set up control "no-phage" competitive mutant fitness assays wherein we replaced phages with simply the phage dilution buffer. The Dub-seq library cultures were grown in the wells of a 48-well microplate (700 μl per well) and grew the microplates in Tecan Infinite F200 readers with orbital shaking and OD600 readings every 15 min for 3–8 hr. After the experiment, survivors were collected, pelleted, and stored at −80 ˚C prior to plasmid DNA extraction. For solid plate-based assays, we incubated the mixture of phage and diluted Dub-seq library at room

temperature for 15 min, and then we plated the mixture on LB agar plates supplemented with appropriate antibiotics and incubated at 37 ˚C overnight. Next day, the resistant colonies were scraped, resuspended in 1ml LB media, and pelleted. Assay pellets were typically stored at −80 ˚C prior to plasmid DNA extraction. We also performed these assays using *E. coli* strains with empty plasmid (used in Dub-seq library creation).

## BarSeq of RB-TnSeq and Dub-seq pooled fitness assay samples

We isolated genomic DNA from RB-TnSeq library samples using the DNeasy Blood and Tissue kit (Qiagen). Plasmid DNA from the Dub-seq library samples was extracted either individually using the Plasmid miniprep kit (Qiagen) or in 96-well format with a QIAprep 96 Turbo miniprep kit (Qiagen). We performed 98 ˚C BarSeq PCR protocol as described previously [64,66] with the following modifications. BarSeq PCR in a 50 μl total volume consisted of 20 μmol of each primer and 150–200 ng of template genomic DNA or plasmid DNA. For the HiSeq4000 runs, we used an equimolar mixture of BarSeq_P2 primers along with new Barseq3_P1 primers. The BarSeq_P2 primer contains the tag that is used for demultiplexing by Illumina software, and the new Barseq3_P1 primer contained an additional sequence to verify that it came from the expected sample. The new Barseq3_P1 primer contains the same sequence as BarSeq_P1 reported earlier with 1–4 N's (which varies with the index) followed by the reverse (not the reverse complement) of the 6-nucleotide index sequence [251]. This modification to earlier BarSeq PCR protocol was done to eliminate the barcode bleed-through problem in sequencing and also to aid in cluster and sample discrimination on the HiSeq4000. All experiments done on the same day and sequenced on the same lane are considered as a "set." Equal volumes (5 μl) of the individual BarSeq PCRs were pooled, and 50 μl of the pooled PCR product was purified with the DNA Clean and Concentrator kit (Zymo Research). The final BarSeq library was eluted in 40 μl water. The BarSeq libraries were sequenced on Illumina HiSeq4000 instrument with 50 SE runs. We usually multiplexed 96 Barseq samples per lane for both RB-TnSeq and Dub-seq library samples.

## Data processing and analysis of BarSeq reads

RB-TnSeq fitness data were analyzed as previously described [64] with additional filters as presented below. Briefly, the fitness value of each strain (an individual transposon mutant) is the normalized log2(strain barcode abundance at end of experiment/strain barcode abundance at the start of the experiment). The fitness value of each gene is the weighted average of the fitness of its strains. We applied filters on experiments such that the mean reads per gene is $\geq$10. As we have reported earlier [64], in a typical experiment without stringent positive selection, gene fitness score (fit) >1 and associated t-like statistic >4 suffices to give a low rate of false positives. Because of the stringent positive selection in phage assays, our standard quality metrics reported earlier [64] were not suitable. Because most of the sequencing reads were from a handful of phage-resistant mutants in the population, the majority of the strains in the library did not have enough reads to accurately calculate fitness values. To determine suitable filters, we compared fitness data between two halves of each gene. As we have several insertions in most genes, we can compute fitness values for the two halves of a gene separately and then plot the first half and second half fitness values for each gene (that has sufficient coverage) against each other as described earlier [64]. These plots are available in the figshare file https://doi.org/ 10.6084/m9.figshare.11413128. Phage T2 assays on solid agar showed poor consistency and were dropped out of the analysis. We observed from the first half and second half fitness plots that fitness values <5 were often not consistent. To reduce false positives, we required that fit $\geq$ 5; t $\geq$ 5; standard error = fit/t $\leq$ 2; and fit $\geq$ maxFit − 8, where maxFit is the highest

fitness value of any gene in an experiment. The limit of maxFit − 8 was chosen based on the fitness score of positive controls (host factors that are known to interfere with phage growth when deleted) and from experimental validations of new hits. The fitness data for all strains in the RB-TnSeq library are provided at https://doi.org/10.6084/m9.figshare.11413128.

For the Dub-seq library, fitness data were analyzed using *Barseq* script from the *Dub-seq* python library with default settings as previously described [66]. From a reference list of barcodes mapped to the genomic regions (BPSeq and BAGseq), and their counts in each sample (BarSeq), we estimate fitness values of each genomic fragment (strain) using *fscore* script from the Dub-seq python library. The *fscore* script identifies a subset of barcodes mapped to the genomic regions that are well represented in the time-zero samples for a given experiment set. We require that a barcode to have at least 10 reads in at least one time-zero (sample before the experiment) sample to be considered a valid barcode for a given experiment set. Then the *fscore* script calculates fitness score (normalized ratio of counts between the treatment sample and sum of counts across all time-zero samples) only for the strains with valid barcodes. The fitness scores calculated for all Dub-seq fragments, we estimate a fitness score for each individual gene *gscore* that is covered by at least one fragment as detailed earlier using nonnegative least squares regression [66]. The nonnegative regression determines if the high fitness of the fragments covering the gene is due to a particular gene or its nearby gene, and avoids overfitting. We applied the same data filters as reported earlier [66] to ensure that the fragments covering the gene had a genuine benefit. Briefly, we identify a subset of the effects to be reliable, if the fitness effect was large relative to the variation between start samples ($|\text{score}| \geq 2$); the fragments containing the gene showed consistent fitness (using a *t* test); and the number of reads for those fragments was sufficient for the gene score to have little noise. As in RB-TnSeq Bar-Seq analysis, Dub-seq data also showed strong positive selection in the presence of phages, where few strains accounting for the most reads and displaying very high fitness scores. To reduce false positives, we applied an additional filter of gene fitness score $\geq 4$ threshold. This threshold cutoff was chosen by analyzing the fragment fitness scores against the number of supported reads and whether the cutoff value agrees with experimental validations carried out in this work. Finally, we estimate the FDR for high-confidence effects as detailed earlier [66]. Briefly, to estimate the number of hits in the absence of genuine biological effects, we randomly shuffled the mapping of barcodes to fragments, recomputed the mean scores for each gene in each experiment, and identified high-confidence effects as for the genuine data. We repeated the shuffle procedure 10 times. The complete dataset of fragment and gene scores are provided at https://doi.org/10.6084/m9.figshare.11838879.v2. Dub-seq data for *E. coli* strain with empty plasmid (used in Dub-seq library creation) were consistent with *E. coli* with no plasmid, indicating we do not see any plasmid-mediated fitness effects. Phage P2 assays were excluded from the *E. coli* K-12 Dub-seq study, as the sequencing runs did not pass the data analysis filters because of low sequencing reads per assay. We used the Dub-seq viewer tool from the *Dub-seq* python library (https://github.com/psnovichkov/DubSeq) to generate regions of the *E. coli* chromosome covering fragments (gene-browser mode). The fitness data for all strains in the Dub-seq library are provided at https://doi.org/10.6084/m9.figshare.11838879.v2.

## *E. coli* MG1655 CRISPRi library

The design and construction of *E. coli* MG1655 CRISPRi library and the overall quality of sgRNAs used are detailed elsewhere [69]. Briefly, the *E. coli* MG1655 CRISPRi library is made up of 32,992 unique sgRNAs targeting 4,457 genes (including small RNA genes, insertion elements, and prophages), 7,442 promoter regions, and 1,060 transcription factor binding sites.

The sgRNA library is driven by pBAD promoter (induced by arabinose) and is expressed from a high copy plasmid (ColE1 replication origin) in *E. coli* K-12 MG1655 strain harboring a genomically encoded aTc-inducible *dCas9* gene.

To perform pooled fitness experiments using *E. coli* MG1655 CRISPRi library, a single aliquot of the library was thawed, inoculated into 5 ml of LB medium supplemented with carbenicillin, kanamycin, and glucose and grown to OD600 of 0.5. At mid-log phase, we collected cell pellets as an initial time point (time-zero) of the library and diluted the remaining culture in induction media (LB broth with arabinose [0.1%], aTc [200 ng/mL], carbenicillin 100 μg/mL, and kanamycin 30 μg/mL) to initiate dCas9 and sgRNA expression for about six doublings (to OD600 about 0.5). Finally, we diluted the library to OD600 of 0.1 in 2X LB supplemented with induction media and incubated 350 μl of the library with 350 μl of diluted phage stocks (MOI of 1) in 5-ml test tubes at room temperature for 15 min. We also set up a control "no-phage" culture wherein we simply mixed the phage dilution buffer with the library. We then moved these competitive fitness assay cultures to 37 ˚C with shaking (200 rpm) for 2 hr, collected the entire cell pellet after 2 hr, and stored at −80 ˚C prior to plasmid DNA extraction to isolate the library. The plasmid library was extracted using QIAprep Spin Miniprep Kit, used for PCR to generate Illumina sequencing samples, and sequenced on Illumina HiSeq2500 instrument with 100 SE runs. Only samples that yielded more than 2 million reads (average library read depth of approximately 60) were used in the downstream analysis.

Illumina reads were quality filtered, trimmed, and mapped using the same procedure as earlier [69] to generate strain abundances (i.e., read counts) for each sgRNA library member (recall that each strain in the library is uniquely determined by the 20-bp variable region of the sgRNA it harbors). The fitness of each sgRNA library member was calculated as the log2 fold-change in abundance of the sgRNA after the experiment versus before the experiment using edgeR [252,253]. For a given enrichment comparison, sgRNAs with fewer than 10 read counts in each replicate of the time-zero and end of experiment samples were filtered out of the analysis. In the edgeR pipeline, each sample was normalized for sequencing depth using the edgeR function calcNormFactors, pseudocounts and dispersions were calculated using the edgeR functions estimateCommonDisp and estimateTagwiseDisp, and the log2 fold-change and corresponding *p*-values were calculated using the edgeR function exactTest, which is based on a quantile-adjusted conditional maximum likelihood (qCML) method. FDR-adjusted *p*-values were calculated using the Benjamini-Hochberg method. We performed a post hoc analysis on these fitness scores and picked a threshold of fitness ≥2 and FDR-adjusted *p*-value <0.05 for new hits of interest based on the satisfaction of these criteria by both positive controls (known phage receptors for example *fadL* for T2 phage and *waa* operon for 186 phage) and validated hits (for example, *dsrB* in the presence of N4 phage). Finally, gene fitness scores were calculated by taking the median fitness scores of (filtered) sgRNAs targeting a given gene. The fitness data for all strains in the CRISPRi library are provided at https://doi.org/10.6084/m9.figshare.11859216.

### Construction of *igaA* mutant strains

We created our *igaA* genetic disruption mutant through a modified recombineering method using pSIM5 [66,254] and approximate insertion site based off of RB-TnSeq mapping (S1 Fig). Primers were designed to amplify a kanR selection marker with 50-bp homology to the insertion site of interest corresponding to a kanR insertion at 51 bp internal to *igaA*. PCRs were generated and gel-purified through standard molecular biology techniques [247] and stored at −20 ˚C until use. Deletions were performed by incorporating the above dsDNA template into the BW25113 genome through standard pSIM5-mediated recombineering methods. First, a

temperature-sensitive recombineering vector, pSIM5, was introduced into BW25113 through standard electroporation protocols and grown with chloramphenicol at 30 ˚C. Recombination was performed through electroporation with an adapted pSIM5 recombineering protocol. Post recombination, clonal isolates were streaked onto kanamycin plates without chloramphenicol at 37 ˚C to cure the strain of pSIM5 vector, outgrown at 37 ˚C, and stored at −80 ˚C until use. A detailed protocol is available upon request. Disruptions were verified by colony PCR followed by Sanger sequencing at the targeted locus and 16S regions. The *igaA* mutant map is given at https://benchling.com/s/seq-pZEspfGx8K9tYzixzMoJ.

## Experimental validation of individual phage resistance phenotypes

To validate the phage resistance phenotypes from both LOF and GOF screens, we performed EOP assays on select *E. coli* mutants from the Keio collection [105] and overexpression ASKA library [105,161], respectively. The complete list of primers, plasmids, and strains used in validation assays are provided in S10–S12 Tables. All deletion strains and plasmids used in this work were confirmed via Sanger sequencing. An isogenic deletion mutant in *E. coli* BL21 for *fadL* was generated by phage P1–mediated transduction of kanamycin resistance from individual Keio mutants [105]. The gene deletion strains from Keio library and the BL21 transductants were cultured in LB media or LB agar supplemented with kanamycin (25 μg/ml), whereas ASKA strains were cultured in LB media or LB agar supplemented with chloramphenicol (30 μg/ml). The plasmids from ASKA library were extracted using QIAprep Miniprep kit (Qiagen), and electroporated into *E. coli* BW25113 strains for GOF validations or respective Keio mutants for complementation of LOF phenotypes. The bottom agar was supplemented with 0.1 mM IPTG to induce protein production from ASKA plasmids. Based on toxicity associated with gene overexpression, IPTG levels (0–0.1 mM) were adjusted and cultures were grown overnight at 30 ˚C.

Phages were quantified by spot titration method. Two microliters of serially 10-fold diluted phages were spotted on a solidified lawn of approximately 4 ml 0.5% top agar inoculated with 100 μl of a fresh overnight bacterial culture and incubated overnight at either 30 ˚C or 37 ˚C. The EOP was calculated as the ratio of the number of plaques on mutant or overexpression strain to the number of plaques on the parental strains (BW25113 or BL21). The EOPs were calculated at least by two biological replicates. S5 Fig lists all EOP estimation numbers and plaque images.

## RNA-seq experiments

Transcriptomes were collected and analyzed for strains BW25113 ($N$ = 3), knockout mutants for *igaA* ($N$ = 3), and overexpression strains for *pdeB* ($N$ = 1), *pdeC* ($N$ = 1), *pdeL* ($N$ = 3), *pdeN* ($N$ = 1), *pdeO* ($N$ = 1), and *ygbE* ($N$ = 3). All cultures for RNA-seq experiments were performed on the same day from unique overnights and subsequent outgrowths.

Strains were grown overnight in LB with an appropriate selection marker at 30 ˚C. Strains were diluted to OD600 approximately 0.02 in 10 mL LB with the appropriate selection marker and, for overexpression strains, 0.1 mM IPTG. Cultures were grown at 30 ˚C at 180 rpm until they reached an OD600 0.3–0.4. Samples were collected as follows: 400 μL of culture was added to 800 μL RNAProtect (Qiagen), incubated for 5 min at room temperature, and centrifuged for 10 min at 5,000*g*. RNA was purified using RNeasy RNA isolation kit (Qiagen) and quantified and quality-assessed by Bioanalyzer. Library preparation was performed by the Functional Genomics Laboratory (FGL), a QB3-Berkeley Core Research Facility at UC Berkeley. Illumina Ribo-Zero rRNA Removal Kit was used to deplete ribosomal RNA. Subsequent library preparation steps of fragmentation, adapter ligation, and cDNA synthesis were

performed on the depleted RNA using the KAPA RNA HyperPrep kit (KK8540). Truncated universal stub adapters were used for ligation, and indexed primers were used during PCR amplification to complete the adapters and to enrich the libraries for adapter-ligated fragments. Samples were checked for quality on an Agilent Fragment Analyzer. Sequencing was performed at the Vincent Coates Sequencing Center on a HiSeq4000 using 100PE runs.

### RNA-seq data analysis

For all RNA-seq experiments, analyses were performed through a combination of KBase [255] and custom jupyter notebook-based methods. Briefly, Illumina reads were trimmed using Trimmomatic [256] v0.36 and assessed for quality using FASTQC. Trimmed reads were mapped to the *E. coli* BW25113 genome (NCBI accession: CP009273) with HISAT2 [257]. Alignments were quality-assessed with BAMQC. From this alignment, transcripts were assembled and abundance-estimated with StringTie [258]. For experiments with sufficient biological replicates, tests for differential expression were performed on normalized gene counts by DESeq2 (negative binomial generalized linear model) [259]. Additional analyses for all experiments were performed in Python3 and visualized employing matplotlib and seaborn packages. For analyses involving DESeq2, conservative thresholds were employed for assessing differentially expressed genes. Genes were considered differentially expressed if they possessed a Bonferroni-corrected *p*-value below a threshold of 0.001 and an absolute log2 fold-change greater than 2. For supplemental analyses without tests for differential expression, data were analyzed by comparing log2 fold-changes of transcripts per kilobase million (TPM) counts. The complete datasets are provided in S6 and S7 Tables.

### Supporting information

**S1 Fig. Map of *igaA* mutants.** *igaA* mutants and their overall fitness scores from *E. coli* K-12 RB-TnSeq screen (S1 Table). Schematic of Rcs phosphorelay with the predicted topology of IgaA [165] is shown at the top. RB-TnSeq mutants in IgaA are mapped to the predicted topology of IgaA (top) and also mapped on to *igaA* nucleotide sequence, with mutant position and fitness scores. Mutants A, B, and D were constructed and their mucoidy and phage resistance phenotype were confirmed (mutant A data are shown in Fig 5A). Though full-length deletion of *igaA* has not been possible, our results indicate that disruption between amino acid 22 and 151 is dispensable. Strain with *igaA* mutant A was further subjected to EOP and RNA-seq analysis presented in the main text. The underlying data for this figure can be found in S1 Data. EOP, efficiency of plating; RB-TnSeq, random barcode transposon site sequencing; RNA-seq, RNA sequencing.
(TIF)

**S2 Fig. Dub-seq viewer plots for high-scoring *E. coli* K-12 BW25113 genomic fragments in the presence of phages.** Following top candidates are shown: high-scoring fragments encoding *micF* and *degP* for CEV1 phage; *aes*, *cpdA*, *malY*, *glk*, *sdiA*, and *gltP* for λ phage cI857. Red lines represent fragments covering highlighted genes completely (start to stop codon), and gray-colored fragments either cover the highlighted gene partially or do not cover the highlighted gene completely. The underlying data for this figure can be found in S1 Data. Dub-seq, dual-barcoded shotgun expression library sequencing.
(TIF)

**S3 Fig. Dub-seq viewer plots for high-scoring *E. coli* K-12 BW25113 genomic fragments in the presence of phages.** Following top candidates are shown: high-scoring fragments encoding *mcrB* for T2 phage; *ompT* and *glgC* for T3 phage; *ompF* for T4 phage; *lit* for T6 phage; *yjcC*

(*pdeC*), *mprA*, *ddpX*, *yhbJ (rapZ)*, *ylaB (pdeB)* for N4 phage. Red lines represent fragments covering highlighted genes completely (start to stop codon), whereas gray-colored fragments either cover the highlighted gene partially or do not cover the highlighted gene completely. The underlying data for this figure can be found in S1 Data. Dub-seq, dual-barcoded shotgun expression library sequencing.
(TIF)

**S4 Fig. Dub-seq viewer plots for high-scoring *E. coli* K-12 BW25113 genomic fragments in the presence of phages.** Following top candidates are shown: high-scoring fragments encoding *rtn(pdeN)*, *yliE (pdeI)*, *gmr (pdeR)*, *dosP (pdeO)*, and *lrhA* for N4 phage; *gltS*, *gltJ*, and *gltP* for 186 phage; *mcrB* and *lit* for LZ4 phage. Red lines represent fragments covering highlighted genes completely (start to stop codon), whereas gray-colored fragments either cover the highlighted gene partially or do not cover the highlighted gene completely. The underlying data for this figure can be found in S1 Data. Dub-seq, dual-barcoded shotgun expression library sequencing.
(TIF)

**S5 Fig. Estimation of EOP with Keio deletion strains and genetic complementation.** (A and B) EOP experiments with Keio strains and ASKA plasmid complementation of respective genes (indicated by +p) in the presence of different phages. We used no IPTG or 0.1 mM IPTG for inducing expression of genes from ASKA plasmid. All experiments were in *E. coli* K-12 BW25113 strain. EOP, efficiency of plating.
(TIF)

**S6 Fig. Examples of candidates that showed high fitness scores in our Dub-seq screen but failed to show such stronger effect in EOP validation experiments.** (A) EOP experiments with ASKA plasmid expressing genes (shown as +gene names) in the presence of different phages. We used no IPTG and 0.1 mM IPTG for inducing expression of genes from ASKA plasmid. We used *E. coli* K-12 BW25113 strain with an empty vector for EOP calculations. (B) Dub-seq viewer plots for high-scoring fragments (red bars) encoding *yedJ*, along with neighboring genes (gray bars) with fitness score in the presence of T4 phage on the y-axis. The Ecocyc operon [171] view for *yedJ* regions is on the bottom. The genomic fragments encoding *yedJ* also encode a small RNA *rseX* (RNA suppressor of extracytoplasmic stress protease) that binds to RNA binding protein Hfq (a global regulator) and specifically targets *ompA* and *ompC* mRNAs. Our library does not have genomic fragments that can resolve *rseX* contribution to fitness independently. Overexpression of RseX has also been shown to increase biofilm formation [260], indicating the role of *yedJ-rseX* locus on resistance to T4 phage and other phages. The underlying data for this figure can be found in S1 Data. Dub-seq, dual-barcoded shotgun expression library sequencing; EOP, efficiency of plating.
(TIF)

**S7 Fig. Dub-seq and RB-TnSeq data uncovers *flhD* upstream loci important in N4 phage infection.** (A) Dub-seq viewer plots for high-scoring fragments (red bars) encoding *flhD* and upstream region along with neighboring genes (gray bars) with fitness score on the y-axis. Overexpression of *flhD* failed to demonstrate strong phage plating defects in our EOP validation experiments (S6A Fig). (B) The Ecocyc operon view of the regulatory region upstream of *flhD* with multiple transcription factor binding sites [171]. (C) Zoom-in view of the upstream region of *flhD* operon with arrows identifying the location of 13 RB-TnSeq insertion mutants that have fitness score >10 in the presence of N4 phage. These results indicate the role of *flhD* regulatory region on N4 phage growth. We speculate that these TnSeq mutants of *flhD* regulatory region may not be overexpressing *flhD*, as our EOP experiments failed to validate *flhD*

overexpression as the cause of N4 phage resistance. The underlying data for this figure can be found in S1 Data. Dub-seq, dual-barcoded shotgun expression library sequencing; RB-TnSeq, random barcode transposon site sequencing.
(TIF)

**S8 Fig. Description of *E. coli* BL21 Dub-seq library.** (A) The fragment insert size distribution in the *E. coli* BL21 Dub-seq library. (B) Cumulative distribution plot showing the percentage of genes in the *E. coli* BL21 genome (y-axis) covered by a number of independent genomic fragments (x-axis). (C) The distribution of the number of genes that are completely covered (start to stop codon) per genomic fragment in the *E. coli* BL21 Dub-seq library. (D) Genome coverage of *E. coli* BL21 Dub-seq library in 10,000-kB windows mapped to *E. coli* BL21-DE3. The underlying data for this figure can be found in S1 Data. Dub-seq, dual-barcoded shotgun expression library sequencing.
(TIF)

**S1 Table. RB-TnSeq *E. coli* K-12 dataset.** RB-TnSeq, random barcode transposon site sequencing.
(XLSX)

**S2 Table. Summary of known and new high fitness score hits per phage per screen at any MOI.** MOI, multiplicity of infection.
(PDF)

**S3 Table. Literature table: Mapping RB-TnSeq, CRISPRi, and Dub-seq hits to reported phage resistance data.** CRISPRi, CRISPR interference; Dub-seq, dual-barcoded shotgun expression library sequencing; RB-TnSeq, random barcode transposon site sequencing.
(PDF)

**S4 Table. CRISPRi *E. coli* K-12 dataset.** CRISPRi, CRISPR interference.
(XLSX)

**S5 Table. Dub-seq *E. coli* K-12 dataset.** Dub-seq, dual-barcoded shotgun expression library sequencing.
(XLSX)

**S6 Table. RNA-seq dataset.** RNA-seq, RNA sequencing.
(XLSX)

**S7 Table. RNA-seq DESeq2_Summary.** RNA-seq, RNA sequencing.
(XLSX)

**S8 Table. RB-TnSeq *E. coli* BL21 dataset.** RB-TnSeq, random barcode transposon site sequencing.
(XLSX)

**S9 Table. Dub-seq *E. coli* BL21 dataset.** Dub-seq, dual-barcoded shotgun expression library sequencing.
(XLSX)

**S10 Table. List of primers.**
(XLSX)

**S11 Table. List of plasmids.**
(XLSX)

**S12 Table. List of strains: Phage and bacteria.**
(XLSX)

**S13 Table. MOI per phage per assay.** MOI, multiplicity of infection.
(XLSX)

**S1 Text. Detailed information on host factors important in phage infection and top-scoring hits for each screen per phage.**
(DOCX)

**S1 Data. The underlying data for all figures.**
(XLSX)

## Acknowledgments

Authors would like to thank Studier F. William, Lucia B. Rothman-Denes, Ian J. Molineux, Jason J. Gill, and Paul Turner for sharing reagents and helpful discussions.

## Author Contributions

**Conceptualization:** Vivek K. Mutalik, Adam P. Arkin.

**Data curation:** Vivek K. Mutalik, Pavel S. Novichkov, Morgan N. Price, Adam P. Arkin.

**Formal analysis:** Vivek K. Mutalik, Benjamin A. Adler, Harneet S. Rishi, Pavel S. Novichkov, Morgan N. Price, Adam M. Deutschbauer, Adam P. Arkin.

**Funding acquisition:** Vivek K. Mutalik, Adam P. Arkin.

**Investigation:** Vivek K. Mutalik, Benjamin A. Adler, Harneet S. Rishi, Denish Piya, Crystal Zhong, Britt Koskella, Pavel S. Novichkov, Morgan N. Price, Adam M. Deutschbauer, Adam P. Arkin.

**Methodology:** Vivek K. Mutalik, Benjamin A. Adler, Harneet S. Rishi, Denish Piya, Britt Koskella, Elizabeth M. Kutter, Pavel S. Novichkov, Morgan N. Price, Adam M. Deutschbauer, Adam P. Arkin.

**Project administration:** Vivek K. Mutalik, Adam P. Arkin.

**Resources:** Vivek K. Mutalik, Harneet S. Rishi, Elizabeth M. Kutter, Richard Calendar, Adam M. Deutschbauer.

**Software:** Pavel S. Novichkov, Morgan N. Price.

**Supervision:** Vivek K. Mutalik, Adam P. Arkin.

**Validation:** Vivek K. Mutalik, Benjamin A. Adler, Denish Piya, Adam M. Deutschbauer.

**Visualization:** Vivek K. Mutalik, Benjamin A. Adler, Adam P. Arkin.

**Writing – original draft:** Vivek K. Mutalik.

**Writing – review & editing:** Vivek K. Mutalik, Benjamin A. Adler, Harneet S. Rishi, Denish Piya, Britt Koskella, Richard Calendar, Morgan N. Price, Adam M. Deutschbauer, Adam P. Arkin.

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
