## [Editor Report · Decision Letter 0]

16 Apr 2020

Dear Dr Mutalik, 

Thank you for submitting your manuscript entitled "High-throughput mapping of the phage resistance landscape in E. coli" for consideration as a Research Article by PLOS Biology.

Your manuscript has now been evaluated by the PLOS Biology editorial staff, as well as by an academic editor with relevant expertise, and I'm writing to let you know that we would like to send your submission out for external peer review.

Please re-submit your manuscript within two working days, i.e. by Apr 20 2020 11:59PM.

Kind regards,

Roli Roberts

Senior Editor

PLOS Biology

---

## [Decision Letter · Decision Letter 1]

24 May 2020

Dear Dr Mutalik,

Thank you very much for submitting your manuscript "High-throughput mapping of the phage resistance landscape in E. coli" for consideration as a Research Article at PLOS Biology. Your manuscript has been evaluated by the PLOS Biology editors, an Academic Editor with relevant expertise, and by three independent reviewers.

You'll see that while the reviewers are broadly positive about your study, they each raise a few concerns that should be addressed. The Academic Editor asked me to emphasise that reviewer #1 asks for a few important clarifications (including why affected RM mechanisms don't show more clear resistance phenotypes), and reviewers #2 and #3 request several additional essential pieces of information (e.g. quality of the GoF/LoF libraries, MOI and host/phage densities in the assays); we also anticipate that the requests by reviewer #3 to reduce the text and improve readability will involve significant work (his recommendations seem sensible, especially given our wide readership).

In light of the reviews (below), we are pleased to offer you the opportunity to address the comments from the reviewers in a revised version that we anticipate should not take you very long. We will then assess your revised manuscript and your response to the reviewers' comments and we may consult the reviewers again.

We expect to receive your revised manuscript within 1 month.

**IMPORTANT - SUBMITTING YOUR REVISION**

*Resubmission Checklist*

*Published Peer Review*

*PLOS Data Policy*

*Blot and Gel Data Policy*

Sincerely,

Roli Roberts

Senior Editor

PLOS Biology

REVIEWERS' COMMENTS:

Reviewer #1:

Overview: The resistance strategies used by E. coli to prevent bacteriophage infection have previously been characterized for several phage:bacteria pairs by using labor-intensive methods that are difficult to scale. To develop an unbiased and comprehensive screen compatible with many bacterial and phage strains, Mutalik and colleagues utilized three previously published E. coli libraries, two for loss of function (RbTn-Seq and CRISPRi) and one for gain of function (DubSeq). With these libraries in two strains of E. coli and 14 diverse phages across three families, the authors were able to validate many existing phage receptors and uncover unexpected pathways involved in phage infection, including colonic acid synthesis and cyclic-di-GMP. Impressively, many hits were validated through plaque assays, and RNA-sequencing was performed to further determine mechanism for previously undescribed hits. 

All in all, this paper is an incredible amount of work, that I believe will be of interest to a broad readership as it opens up a lot of avenues for future research and adaptation to other phage pairs. Before I can fully recommend for publication, I only have a few concerns to address: 

Major points:

* The mention of resistance mechanisms throughout the introduction primes the reader to think about classic bacteria defense mechanisms such as Restriction/Modification, Methylation, CRISPR etc. Besides CRISPR, which is not expressed in these strains, these are all likely categories of genes that would appear as hits in the DubSeq, overexpression experiments. Why do the authors think that restriction or methylation enzymes are not strong hits? Along these lines, the conclusion section would benefit from a further discussion into the conditions under which a bacterium would actively overexpress the colanic acid synthesis pathway as a phage-defense mechanism. 

* Most of the phage strains in the study are lytic, as suggested by the choice of using lambda cI857 and p1vir. However, P2 and 186 are known to be lysogenic phages. Whether lytic phages were targeted in this study, and how the lysogenic nature of P2 and 186 could have affected the results is not addressed. Could certain bacterial mutants bias lysogeny of those phages leading to "resistance" as indicated by growth of those lysogens? Also, why is phage P2 excluded from the K-12 DubSeq study?

Minor Points:

* Line 156 - "As described later, for comparative purposes, we also performed high-throughput genetic assays in the E. coli BL21 strain background…. Different serotypes of E. coli are also important pathogens with significant global threat and are crucial players in specific human-relevant ecologies [84-86], leading to the question of whether strain variation is also important in predicting the response to phage-mediated selection or whether the mechanisms are likely to be conserved." 

o This study does not address pathogenic strains of E. coli and gives an unclear motivation for the use of strain BL21. The organization of this paragraph guides the reader think that non-pathogenic and pathogenic strains will be compared, whereas this study characterizes two similar non-pathogenic strains. These strains may be expected to behave more similarly than a comparison with a pathogenic strain. Therefore, this paragraph would benefit from a stronger introduction to why strain BL21 is used. 

o The use of two novel shiga-toxin phages in this study is very intriguing and can link to the pathogen point made above. The conclusion could benefit from a discussion of how the genes responsible for preventing infection of these two pathogen associated phages differ from non-pathogenic phages of similar structure. 

* Line 178 - Figure 1. 

o The organization of Figure 1 appears like Myoviridae is used with RB-TnSeq, Podoviridae is used with CRISPRi and Siphoviridae is used with GOF. This figure could benefit from either moving the bacterial strains to "Technology" box and then having one arrow from technology to "Phage Strains" or removing the Bacterial Strain information. 

* Figure 2 - Where are the MOIs described for each phage? The methods section does not describe the MOIs used. 

* Figure 3 - In the CRISPRi screen, igaA is such a strong hit for all phages besides 186. In RbTn-Seq, however, igaA is a strong hit for 186. Why do the authors think that this result was not repeatable with CRISPRi? Does phage 186 contain a depolymerase to degrade the capsule? 

* Line 495 - "This is the first systematic analysis of how gene overexpression impacts phage resistance"

o This is overstated given that Qimron 2006 use the entire ASKA collection to probe T7 resistance. 

* Lines 1010 and 1035 - Should this should be -80°C? 

* Supplementary Figure S5 - There are no 0.1mM IPTG pictures for T5 and T6 phages 

Reviewers #2: 

[identify themselves as Sankar Adhya and Shayla Hesse]

Mutalik et al apply the array of specific and tractable genome-wide perturbations present in RB-TnSeq, Dub-Seq, and CRISPRi E. coli libraries to broadly probe the phage resistance landscape for multiple phage-bacteria pairs. Their work highlights the key advantages of such an approach: overcoming the limitations of complete LOF screens and venturing into the largely uncharted territory of GOF and intergenic screens - all while maintaining a relatively high degree of technical efficiency. Their work reveals novel insights on an age-old subject (coliphage-host interaction and evolution) and is well-presented in both written word and graphics. The degree to which non-phage-receptor- related resistance genes varied for the 14 different phages was particularly interesting, as well as the central role of Rcs signalling (igaA downregulation and rcsA upregulation) in modulating bacterial sensitivity to phage infection more generally. 

I recommend publication of the manuscript after authors satisfactorily respond to some comments listed below.

Comments:

A RB-TnSeq E.coli library seems fairly straightforward, but I'm sure that CRISPRi and Dub-Seq library generation are trickier propositions. One cannot easily assess the overall quality of sgRNAs (In terms of target affinity and avoiding off-target effects). Can you control gene dosage in Dub-Seq library generation? If not, what is the range of variability for different plasmids?

Specific Comments

Strain-specific differences between K-12 and BL21: Side-by-side comparison of Fig. 2a and 7 reveal some notable differences in RB-TnSeq results, while there were many more that were observed with the Dub-Seq data - do you think most of them are artifacts? What do you think are the most likely explanations?

Dub-Seq: What is the average increase in gene copy number on a per cell basis?

Figure 5: consider moving to Supplemental, not sure how much visualization of those raw data enhances the reader's understanding of the key points of the paper

Page 22, Lines 850-855: The authors dutifully point out the potential for phage-resistant strains with significantly lower fitness to be outcompeted to the point of being obscured. It may also be worth pointing out that the relative fitness of these lab-generated strains is distinct from the likelihood that their cognate mutant will naturally evolve under phage selection pressure. (A more obvious limitation, yes, but also more consequential. And particularly relevant given the authors' claim in the abstract that an upshot to the application of their methods may be the generation of "datasets that allow predictive models of how phage-mediated selection will shape bacterial phenotype and evolution.")

Page 30, Line 183-184: The igaA mutant map weblink returned an error message.

Supplementary table S12: recommend addition of phage genome accession numbers/links for the non-canonical phages

Reviewer #3:

[identifies himself as Jeremy J Barr]

The manuscript by Mutalik et. al., presents a tour de force in mapping the phage resistance landscape across two E. coli strains. Using a combination of LOF (Tn-Seq & CRISPRi) and GOF (DubSeq) techniques, they screened for host factors associated with phage resistance across 14 E. coli phages, many of which are well established model phages, along with novel and less-well characterised phages, demonstrating the broad applicability of the technique. Using these techniques, they confirm established phage-receptor mutants, while also uncovering novel resistance mechanisms, some of which were validated using conventional knock-out and complementation. 

The amount of work invested into this manuscript is both impressive and staggering, and this could have easily been split across multiple manuscripts. Yet the amount of data and analysis presented did make parts of the manuscript very detailed and difficult to read, with parts that could be reduced to improve readability. Overall, I have no major concerns with the experimental approach and analysis conducted and believe this is a very important paper to the field. Most of my comments address readability and accessibility of the manuscript. 

I had two major comments with the paper that I felt should be addressed.

1) MOIs and both host and phage concentrations for these most experiments are not reported in text or methods. The authors simply state 'varying MOIs' and provide a qualitative scale for this (with exception of CRISPRi where MOI 1 was used). It is important that both the MOIs and concentrations used in each assay are report to confirm their validity. Please report all MOIs along with actual concentrations of phage and host used.

2) I felt that the authors ignored the limitations of their approach, particularly the time and costs associated with establishing these libraries coupled with the limited phage that may target these library strains. Suggestion is to add an extra section to the discussion on this, and speculating how these limitations may be overcome in future developments of the technology.

3) At points the manuscript was overly descriptive and long. The authors should consider reducing the amount of text in sections to improve readability and focus of each section, some specific examples are below.

See below for line by line comments and suggestions:

Lines 154-159 & 164-166 - Sections describing the libraries and hosts were repetitive, please combine points/sections

Lines 171-172 - Please highlight in text the novel and STEC targeting phages

Line 188 - term 'are' please change to past tense

Lines 194-200 - the descriptions of positive fitness scores were confusingly written and long, I understand the rationale, but recommend simplifying this section

Lines 228-229 - See comment above, please report both your MOIs and phage/host concentrations used.

Line 236 - should read 'more than one'

Lines 243-244 - This section reads like either the LPS or proteinaceous receptors are required for phage infection, this may be the case for your 14 phages studied, but not as a general rule - please rewrite to highlight the large diversity of potential phage receptors 

Figure 2 - Recommend adding darker border lines between phages as it is difficult to interpret inner parts of the graph. Also recommend adding something to the figure panel C to highlight that phages affected are colored purple.

Lines 363 - briefly re-iterate what a high fitness score implies in this assay

Lines 446 - Is not clear where this genomic DNA comes from, assuming this is taken from the same K-12 host, but this should be briefly explained here.

Lines 453 - "67 genome wide GOF competition" - Can you clarify what this means? Does one genome-wide GOF assay encompass a single library containing the whole K12 genome of sheared 3kb DNA?

Lines 457-458 - Again re-iterate what positive-growth implies in this experiment - multiple other points in manuscript this could be explained (lines 517, 

Lines 485-488 - Further explanation needed here, does the increased expression of PDEs lead to degradation of cyclic-di-GMP and thereby inhibition of biofilm formation? How does this confer broad resistance to phages?

Lines 550-561 - reduce complexity here

Lines 566-567 ¬- regarding Rcs pathway, useful to also state what this pathway activates (e.g., colonic acid)

Lines 601-637 - In general this section was difficult to read and a lot of detail could be cut to focus on the main outcomes

Section starting 638 - This section seems entirely focused on N4 phage infectivity - which is needed, but the heading doesnt convey this. please reword

Figure 6 - Panel B: could briefly state the mechanism of igaA knock down and how it leads to colanic acid and EPS activation. Panel D: Similar comment, briefly explain mechanism of up and down regulation

Lines 775-776 - include brief description on which host library was made from - assuming BL21

Figure 7 - Why not highlight yellow boxes for confirmed phage receptors again?

---

## [Decision Letter · Decision Letter 2]

6 Aug 2020

Dear Dr Mutalik,

Thank you for submitting your revised Research Article entitled "High-throughput mapping of the phage resistance landscape in E. coli" for publication in PLOS Biology. I have now obtained advice from the original reviewers and have discussed their comments with the Academic Editor. 

We're delighted to let you know that we're now editorially satisfied with your manuscript. However before we can formally accept your paper and consider it "in press", we also need to ensure that your article conforms to our guidelines. A member of our team will be in touch shortly with a set of requests. As we can't proceed until these requirements are met, your swift response will help prevent delays to publication. Please also make sure to address the Data Policy-related requests noted at the end of this email.

*Copyediting*

*Published Peer Review History*

*Early Version*

*Submitting Your Revision*

Sincerely,

Roli Roberts

Senior Editor,

rroberts@plos.org,

PLOS Biology

DATA POLICY:

Many thanks for depositing your raw data in the SRA and Figshare. However, we also ask that all individual quantitative observations that underlie the data summarized in the figures and results of your paper be made available in one of the following forms:

Regardless of the method selected, please ensure that you provide the individual numerical values that underlie the summary data displayed in the following figure panels as they are essential for readers to assess your analysis and to reproduce it: Figs 2A, 3ABC, 4A, 6ACE, S8. NOTE: the numerical data provided should include all replicates AND the way in which the plotted mean and errors were derived (it should not present only the mean/average values).

REVIEWERS' COMMENTS:

Reviewer #1:

The authors have adequately addressed each of my major and minor comments. I have no additional comments. 

Reviewer #2:

[identifies herself as Shayla Hesse]

Thank you for your clear, well-reasoned, and very thorough responses to each point raised by the reviewers. In our opinion, the revisions made to the manuscript adequately address the issues raised and significantly strengthen the paper. 

Reviewer #3:

[identifies himself as Jeremy Barr]

I would like to thank the authors for their detailed reviewer response and additional work on the revised manuscript. I think this new manuscript is much improved, including some excellent additions, particularly the critical review of the costings and benefits of the GOF & LOF approaches. The authors have addressed all my concerns and I have no further comments,

---

## [Editor Report · Decision Letter 3]

8 Sep 2020

Dear Dr Mutalik,

On behalf of my colleagues and the Academic Editor, J. Arjan G. M. de Visser, I am pleased to inform you that we will be delighted to publish your Research Article in PLOS Biology. 

Early Version

PRESS 

Kind regards,

Alice Musson

Publishing Editor, 

PLOS Biology

on behalf of

Roland Roberts,

Senior Editor

PLOS Biology